# LOX-1 acts as an N⁶-methyladenosine-regulated receptor for *Helicobacter pylori* by binding to the bacterial catalase

Judeng Zeng [1,2,3,17], Chuan Xie [1,2,4,17], Ziheng Huang [1,2,3], Chi H. Cho[5], Hung Chan[1,2,3], Qing Li[1,2,3], Hassan Ashktorab[6,7,8], Duane T. Smoot[9], Sunny H. Wong[10], Jun Yu [1,3,11], Wei Gong[12,13], Cong Liang[14], Hongzhi Xu[15], Huarong Chen [1,2,3], Xiaodong Liu[2,3], Justin C. Y. Wu[1,11], Margaret Ip [3,16], Tony Gin [2], Lin Zhang [1,2,3,11] ✉, Matthew T. V. Chan [2,3] ✉, Wei Hu [12,13] ✉ & William K. K. Wu [1,2,3] ✉

The role of N⁶-methyladenosine (m⁶A) modification of host mRNA during bacterial infection is unclear. Here, we show that *Helicobacter pylori* infection upregulates host m⁶A methylases and increases m⁶A levels in gastric epithelial cells. Reducing m⁶A methylase activity via hemizygotic deletion of methylase-encoding gene *Mettl3* in mice, or via small interfering RNAs targeting m⁶A methylases, enhances *H. pylori* colonization. We identify LOX-1 mRNA as a key m⁶A-regulated target during *H. pylori* infection. m⁶A modification destabilizes LOX-1 mRNA and reduces LOX-1 protein levels. LOX-1 acts as a membrane receptor for *H. pylori* catalase and contributes to bacterial adhesion. Pharmacological inhibition of LOX-1, or genetic ablation of *Lox-1*, reduces *H. pylori* colonization. Moreover, deletion of the bacterial catalase gene decreases adhesion of *H. pylori* to human gastric sections. Our results indicate that m⁶A modification of host LOX-1 mRNA contributes to protection against *H. pylori* infection by downregulating LOX-1 and thus reducing *H. pylori* adhesion.

*Helicobacter pylori* is a microaerophilic Gram-negative pathogen that colonizes the stomach of over 50% of the world population, with a higher prevalence in countries with lower socio-economic status[1,2]. Chronic infection with this pathogen is a predominant driver for a broad range of clinical sequelae in susceptible individuals, including chronic gastritis, peptic ulcer, mucosa-associated lymphoid tissue lymphoma, and gastric adenocarcinoma[3]. The infection is usually acquired in childhood and persists lifelong, unless eradicated by antibiotics-based treatments[4]. According to the consensus of the 2015 Kyoto conference, all *H. pylori*-positive individuals should receive eradication therapy, unless there are competing reasons[5]. However, the eradication of this pathogen has been increasingly difficult owing to the globally increasing antibiotic resistance over the last few decades, with the treatment failure rates now reaching 10 to 30%[6,7]. Since

2017, *H. pylori* has been listed by the World Health Organization as a high-priority pathogen posing serious threat to the human health[8]. Thus, a better understanding of the fundamental mechanism underlying *H. pylori* infection as well as developing novel treatment strategies are urgently needed.

N⁶-methyladenosine (m⁶A) RNA methylation is the most prevalent internal messenger RNA (mRNA) modification in eukaryotes, occurring in approximately 0.1–0.4% of adenosines and predominantly located in 3′-untranslated regions (3′-UTRs), near stop codons, and in terminal exons of over 25% of human transcripts[9]. Owing to the rapid advancement of m⁶A-mapping methods, m⁶A modification is now recognized as a pivotal regulatory mechanism controlling gene expression[10]. m⁶A RNA methylation is a reversible and dynamic process, finely tuned by the opposing activities of m⁶A methyltransferases

(also known as "writers", including methyltransferase-like 3 (METTL3), methyltransferase-like 14 (METTL14), and Wilms tumor 1 associated protein (WTAP)) and m6A demethylases (also known as "erasers", including fat-mass and obesity-associated protein (FTO) or alkylation repair homolog protein 5 (ALKBH5))[11]. To execute the biological functions of m6A modification, m6A-modified mRNAs have to be "read" by YT521-B homology (YTH) domain-containing proteins, including YTHDF1-3, YTHDC1 and YTHDC2 (also known as "readers"), which regulate RNA stability, alterative splicing, nuclear export, and translational efficiency[12]. Prevailing evidence indicates that m6A modification is key to the regulation of diverse physiological processes, including cell differentiation, inflammation, autophagy, and viral infection[13–16]. However, the role of m6A modification in *H. pylori* infection has not been investigated.

In the current study, we discovered that m6A modification was increased in *H. pylori*-infected gastric epithelium, due to upregulation of m6A "writers". Such elevation in m6A level then restricted the colonization of *H. pylori*. Moreover, by an integrative analysis of m6A-sequencing (m6A-seq) and RNA-sequencing (RNA-seq), we identified Lectin-Like Oxidized Low-Density Lipoprotein Receptor-1 (LOX-1) as the direct downstream target of m6A modification.

## Results

### *H. pylori* infection up-regulated cellular m6A "writers" and m6A level

To delineate the interactions between *H. pylori* infection and cellular m6A modifications, we first determined the expression levels of principal m6A "writers" in *H. pylori*-positive and -negative human gastric specimens (Supplementary Table 1 for patients' demographic data). Immunohistochemical (IHC) staining revealed that the expression levels of METTL3, METTL14 and WTAP were all significantly upregulated in *H. pylori*-positive samples when compared to the *H. pylori*-negative specimens. In line with the upregulation of m6A "writers", cellular m6A levels were also increased in *H. pylori*-positive samples (Fig. 1a). Given m6A modification is a reversible process, we determined gastric m6A levels in *H. pylori*-infected patients before and after *H. pylori* eradication with antibiotics. The m6A dot-blot assay showed that the total m6A levels were decreased in 10 out of 12 pairs of patients' specimens after receiving antibiotic treatment (Supplementary Fig. 1; Supplementary Table 2 for patients' demographic data). In the next step, we orally inoculated C57BL/6 mice with *H. pylori* Sydney strain 1 (SS1) for 3 months and the mouse stomachs were collected to examine the expression levels of major m6A "writers" and "erasers". Similar to the human gastric specimens, Mettl3, Mettl14 and Wtap but not Fto or Alkbh5 were markedly increased in *H. pylori*-infected mouse samples (Fig. 1b), accompanied by elevated cellular m6A levels as measured by the m6A dot blot assay ($n = 13$ per group; 5 samples per group shown in Fig. 1c and 8 additional samples per group shown in Supplementary Fig. 2). For in vitro study, two *H. pylori* strains (TN2GF4 and ATCC 43504) were used to co-culture with the human gastric epithelial cells HFE145[17,18], and the expression levels of major m6A "writers" and "erasers" were determined by Western blots. The results showed that METTL3, METTL14 and WTAP were remarkably upregulated, while FTO and ALKBH5 were slightly increased in HFE145 cells during *H. pylori* infection in a time-dependent manner (Fig. 1d), and the cellular m6A level was also significantly increased after co-culturing with *H. pylori* (Fig. 1e).

### Ablation of m6A writers enhanced whereas overexpression of m6A writers reduced *H. pylori* infection

Next, we speculated whether the change of cellular m6A level was involved in the defense against *H. pylori* infection. To answer this question, we first employed *Mettl3*[+/−] mice to determine whether the hemizygous deletion of the principal m6A "writer" could affect *H. pylori* colonization in vivo. Wild-type and *Mettl3*[+/−] mice were orally

inoculated with *H. pylori* strain SS1 for 3 months, and the bacterial colonization in the mouse stomachs was examined by colony formation assay and immunofluorescence. As shown in Fig. 2a, b, the colonization of *H. pylori* in *Mettl3*[+/−] mouse stomachs was remarkably increased when compared to wild-type mice, suggesting that the downregulation of cellular m6A level could promote *H. pylori* infection. We and other investigators previously reported that *H. pylori* invaded into the gastric epithelial cells could serve as a source for chronic infection[19–23]. The effect of m6A ablation on intracellular colonization of *H. pylori* was then explored in vitro. Two individual siRNAs targeting major m6A "writers" (METTL3, METTL14 and WTAP) were respectively transfected into HFE145 cells to ablate cellular m6A levels (Fig. 2c). The transfected cells were further co-cultured with *H. pylori* for 6 h, followed by incubation with gentamycin for 2 h to kill the extracellular bacteria. The intracellular colonization of *H. pylori* was then examined by *H. pylori*-specific 16S ribosomal DNA (16S rDNA) qPCR, immunofluorescence, and colony formation assay. These experiments collectively showed that knockdown of major m6A "writers" significantly increased the intracellular abundance of *H. pylori* (Fig. 2d–f). Furthermore, we overexpressed these major m6A "writers" (METTL3, METTL14, WTAP) in HFE145 cells to increase the cellular m6A levels (Fig. 2g). Consistently, the intracellular colonization of *H. pylori* was decreased in the m6A "writers"-overexpressing cells as shown by all three quantitative assays, namely, 16S rDNA qPCR, immunofluorescence, and colony formation assay (Fig. 2h–j).

### LOX-1 mRNA was identified as an m6A-modified transcript in *H. pylori* infection

To identify the m6A-modified target(s) which was involved in the modulation of *H. pylori* infection, m6A-seq and RNA-seq analysis were performed in HFE145 cells with or without *H. pylori* infection. Results showed that m6A modification of the transcript of Lectin-Like Oxidized Low-Density Lipoprotein Receptor 1 (LOX-1), also known as Oxidized Low Density Lipoprotein Receptor 1 (OLR1), was highly upregulated while the mRNA level of LOX-1 was significantly downregulated upon *H. pylori* infection (Supplementary Figs. 3 and 4). We next examined the data of m6A-seq and RNA-seq in *H. pylori*-infected HFE145 cells with or without WTAP knockdown. Integrative analysis of m6A-seq and RNA-seq showed that the m6A level of LOX-1 was remarkably decreased accompanied by an increase in mRNA level upon knockdown of WTAP in *H. pylori*-infected HFE145 cells (Fig. 3a). The distribution of m6A peaks on LOX-1 mRNA was further visualized by the Integrative Genomics Viewer (IGV) tool, which revealed the marked enrichment of m6A in the 3′-UTR of LOX-1 mRNA upon *H. pylori* infection whereas WTAP knockdown markedly reduced the m6A levels and increased the mRNA level of LOX-1 (Fig. 3b). As reported, LOX-1 serves as a scavenger receptor that mediates oxidized low-density lipoprotein internalization through receptor-mediated endocytosis[24]. Besides, LOX-1 could act as a receptor that promotes the adhesion of *Escherichia coli* and *Mycobacterium tuberculosis* to macrophages and mediates the bacterial internalization[25,26]. Therefore, we speculated that LOX-1 might be an m6A-regulated target that modulated *H. pylori* attachment and invasion into gastric epithelial cells. The m6A modification and transcriptional alteration of LOX-1 were next validated by m6A-qRT-PCR and qRT-PCR, respectively, and the related protein alteration of LOX1 was further confirmed by Western blots (Fig. 3c–e). In addition, the mRNA and protein levels of LOX-1 were upregulated upon knockdown of METTL3, METTL14 or WTAP, confirming that LOX-1 expression was negatively regulated by m6A modification (Fig. 3f, g). Moreover, RNA decay assay showed that knockdown of METTL3, METTL14 or WTAP increased the mRNA stability of LOX-1 (Fig. 3h). To further

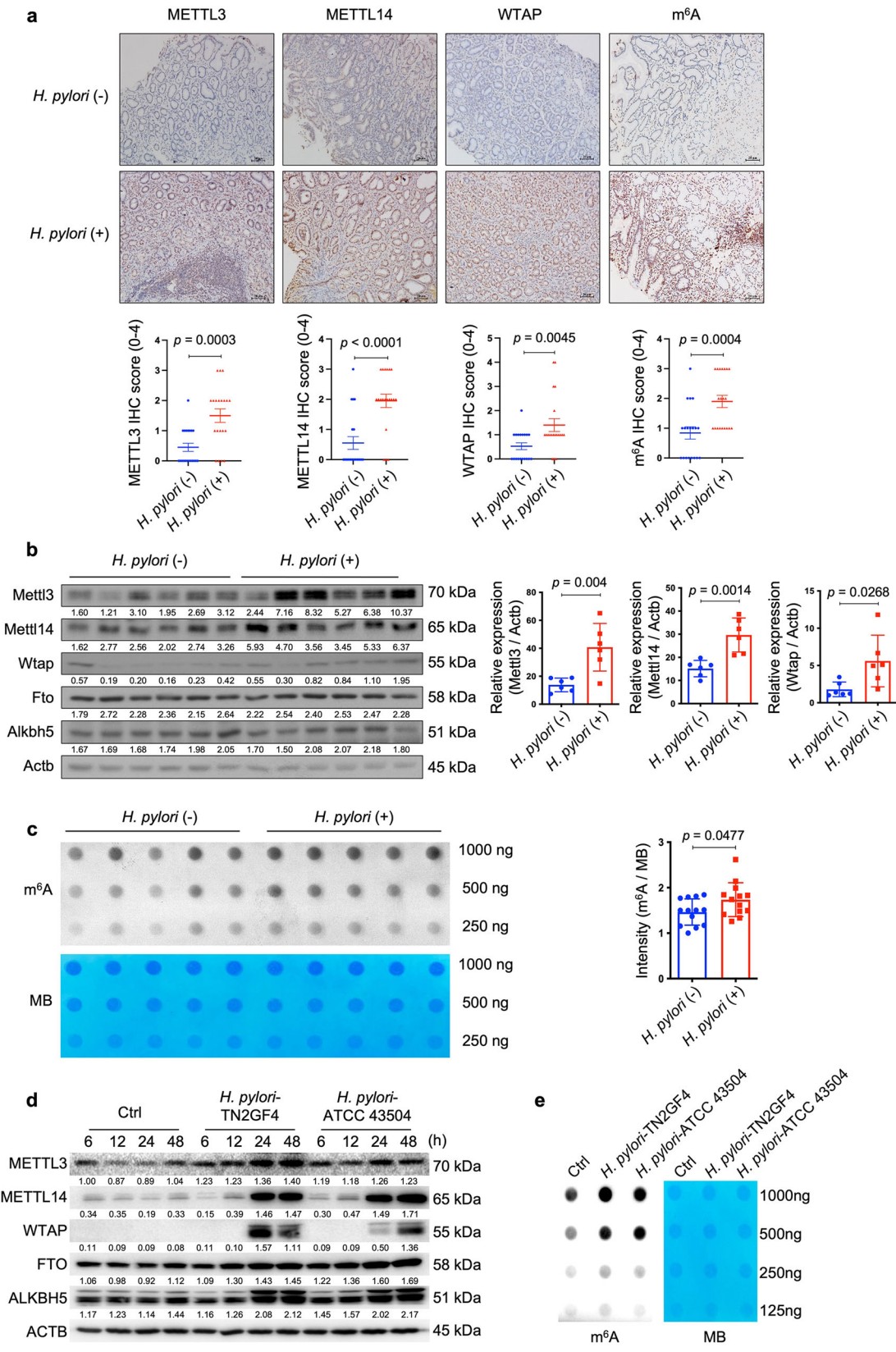

confirm LOX-1 was regulated by m⁶A modification on its 3′-UTR, dual luciferase assay was performed using LOX-1 3′-UTR with or without mutation of the m⁶A sites. As expected, knockdown of m⁶A "writers" upregulated the luciferase activity, whereas mutation of m⁶A sites on the LOX-1 3′-UTR abolished such effects (Fig. 3i).

**LOX-1 served as a cell surface protein to promote the adhesion and invasion of *H. pylori* into gastric epithelial cells by binding to the bacteria catalase**

To delineate the function of LOX-1 in *H. pylori* infection, we first used two individual siRNAs to knock down LOX-1 in HFE145 cells (Fig. 4a). Knockdown of LOX-1 significantly reduced the intracellular

**Fig. 1 | *H. pylori* infection up-regulated the expression levels of METTL3, METTL14, WTAP and increased cellular m⁶A levels in vivo and in vitro.**
**a** Representative images of immunohistochemical (IHC) staining for METTL3, METTL14, WTAP and cellular m⁶A levels in *H. pylori*-positive (*n* = 20 samples) and -negative (*n* = 20 samples) human gastric specimens. Quantitative analysis was shown as means ± SD. Scale bar, 20 μm. **b** Total proteins were extracted from mouse stomach tissues of the control group (*n* = 6 animals) and *H. pylori*-infected group (*n* = 6 animals), and the expression levels of Mettl3, Mettl14, Wtap, Fto and Alknh5 were examined by Western blots. Actb was used as a loading control. The gray scales of each sample were measured by Image J and the quantitative analysis of target/Actb was conducted. Quantitative analysis was shown as means ± SD. **c** The m⁶A level of poly(A)⁺ RNAs in mouse stomach specimens with (*n* = 5 animals) or without (*n* = 5 animals) *H. pylori* colonization was evaluated by m⁶A dot blot assay. The RNA from each sample was loaded equally by a 2-fold serial dilution. MB staining was used as a loading control. The intensity of dots was measured by Image J and quantitative analysis was shown as m⁶A/MB (*n* = 13 per group in total; in

addition to the 5 samples per group shown in this figure, 8 extra samples per group were shown in Supplementary Fig. 2). Quantitative analysis was shown as means ± SD. **d** Human gastric epithelial cells HFE145 were co-cultured with two *H. pylori* strains (TN2GF4, ATCC 43504; MOI = 100). The cells were harvested at different time points (6, 12, 24, 48 h) to measure the protein levels of m⁶A "writers" (METTL3, METTL14, WTAP) and "erasers" (FTO, ALKBH5) by Western blots. ACTB was used as a loading control. The blots shown are representative of three independent experiments with similar results. **e** HFE145 cells were co-cultured with two *H. pylori* strains (TN2GF4, ATCC 43504; MOI = 100) for 24 h. Poly(A)⁺ RNA from total RNA was isolated form *H. pylori*-infected HFE145 cells. The Poly(A)⁺ RNA was loaded onto the membrane with a 2-fold serial dilution. MB staining was used as a loading control. The blots shown are representative of three independent experiments with similar results. Statistical analysis of the data (**a**–**c**) was performed using unpaired two-tailed Student's *t* test and the corresponding *p*-values are included in the figure panels. Source data are provided as a Source Data file.

colonization of *H. pylori* (Fig. 4b–d), whereas overexpressing LOX-1 showed an opposite effect (Fig. 4e–h). Moreover, silencing LOX-1 abolished the effect of WTAP knockdown-mediated increase in bacterial load in HFE145 cells, indicating that LOX-1 was an important downstream effector in m⁶A-regulated *H. pylori* infection (Fig. 4i). To further confirm the functional role of LOX-1 in vivo. Wild-type and *Lox-1⁻/⁻* mice were infected with *H. pylori* for 1 month, and the bacterial colonization in the mouse stomachs was examined by colony formation assay and immunofluorescence. As shown in Fig. 4j–k, *H. pylori* colonization was significantly decreased in *Lox-1⁻/⁻* mice. We also observed the colocalization between Lox-1 and *H. pylori* in wild-type mice while these colocalized signals disappeared in *Lox-1⁻/⁻* mice, suggesting that Lox-1 might serve as a cell surface receptor for *H. pylori* adhesion and colonization. Moreover, hematoxylin and eosin (H&E) staining of mouse gastric specimens was performed, confirming that the inflammatory cell infiltration was reduced in *Lox-1⁻/⁻* mice upon *H. pylori* infection (Supplementary Fig. 5). By expressing GFP-LOX-1 and staining *H. pylori* in HFE145 cells, we also observed the colocalized signals on cell surface in vitro, which further confirmed the interactions between LOX-1 and *H. pylori* (Fig. 5a). Moreover, bacterial adhesion assay was performed, and the results showed that LOX-1 knockdown decreased the adhesion levels of *H. pylori* to HFE145 cells, whereas overexpression of LOX-1 showed an opposite effect, suggesting that LOX-1 could act as a cell membrane protein to support *H. pylori* attachment on gastric epithelial cells (Fig. 5b, c). Next, we sought to identify which outer membrane protein(s) of *H. pylori* could interact with LOX-1. To achieve this, we first co-incubated the *H. pylori*-extracted proteins with HFE145 cell lysates. Then, specific LOX-1 antibody was used to pull down LOX-1 as well as its interacted proteins from the mixture, and the enriched proteins were further separated on SDS-PAGE followed by silver staining (Fig. 5d). A specific protein band above the IgG heavy chain was identified as the catalase of *H. pylori* by mass spectrometry analysis (Fig. 5d). Furthermore, we performed reciprocal coimmunoprecipitation to validate the direct interaction between LOX-1 and *H. pylori* catalase. As expected, we confirmed the direct binding between LOX-1 and catalase (Fig. 5e).

It has been reported that catalase was located on the bacterial surface during autolysis of *H. pylori*[27]. To confirm the involvement of catalase in LOX-1-mediated *H. pylori* adhesion, we constructed a recombinant His-tagged catalase for generating catalase-coated fluorescent beads (Supplementary Fig. 6). To simulate bacterial infection, the coated beads were incubated with HFE145 cells in a multiplicity of infection (MOI) of 1:100. The confocal data showed that catalase coating significantly promoted the adhesion and internalization of fluorescent beads in gastric epithelial cells (Fig. 5f). Moreover, knockdown of LOX-1 reduced the adhesion and internalization of catalase-coated beads, whereas overexpression of LOX-1 produced an opposite effect (Fig. 5g, h). To further confirm the

bacterial catalase was indeed involved in the adhesion of *H. pylori* to human gastric tissues, we next constructed an isogenic catalase-negative mutant with ATCC 43504 (*ΔkatA*) using an allelic exchange method to inactivate the catalase-coding gene *katA*. Genetic complementation of *katA* in *ΔkatA* mutant (*ΔkatA* + *katA*) was also performed (Supplementary Figs. 7 and 8). To further confirm if *katA* knockout could only affect catalase expression but not other proteins of *H. pylori*, silver staining was performed to analyze the whole-cell bacterial proteins. A protein band slightly above the 55-kDa marker (the molecular weight of catalase is 58 kDa) disappeared in both soluble and insoluble protein fractions of *ΔkatA* strain but no obvious changes of other protein bands could be observed (Supplementary Fig. 9a). Moreover, qRT-PCR was performed to detect the mRNA expression of two major *H. pylori* adhesin-encoding genes (i.e., *babA* and *sabA*) in wild-type, *ΔkatA* and *ΔkatA* + *katA* strains. The results showed *KatA* knockout or its genetic supplementation did not affect *babA* or *sabA* expression (Supplementary Fig. 9b). Next, by incubating fluorescein isothiocyanate (FITC)-labeled wild-type ATCC 43504 or its *ΔkatA* or *ΔkatA* + *katA* strains with sections of human normal gastric tissue array (72 section cores from 24 cases (i.e., triplicate sections from each case); US Biomax, BN01011B), we found that the adhesion levels of catalase-negative mutant strain were markedly decreased in gastric body (*n* = 11, range and mean of inhibition: 11.4–92.0% and 48.8%) and antrum tissues (*n* = 8, range and mean of inhibition: 7.4–60.7% and 44.2%), and the adhesion was restored after genetic complementation of *katA* gene (gastric body: range and mean of restoration: 40.9–144.4% and 80.9%; gastric antrum: range and mean of restoration: 8.3–142.9% and 49.3%; with 100% restoration corresponds to the complete restoration the adhesion level back to that of the wild-type strain). However, no significant changes were observed in cardia tissues (*n* = 5, range and mean of inhibition: −9.9–41.1% and 18.8%; range and mean of restoration: −28.6–328.6% and 86.3%; Fig. 5i, Supplementary Fig. 10). Considering pre-heating of the tissue array (to avoid tissue detachment) at 60 °C before the staining process might denature tissue proteins, including the putative receptor(s) for bacterial adhesion, we also compared the adhesion level of the three *H. pylori* strains (i.e., wild-type, *ΔkatA* and *ΔkatA* + *katA* strains) in human gastric sections without or with pre-heating. The results showed that pre-heating at 60 °C slightly reduced *H. pylori* binding to the human gastric tissues but did not alter the inhibitory effect of *katA* knockout or the restoring effect of *katA* genetic complementation (Supplementary Fig. 11).

## Blockage of LOX-1 by a small molecule inhibitor suppressed *H. pylori* infection in vitro and in vivo
A newly identified small-molecule inhibitor of LOX-1, referred to as BI-0115, was reported to selectively target LOX-1 and potently block

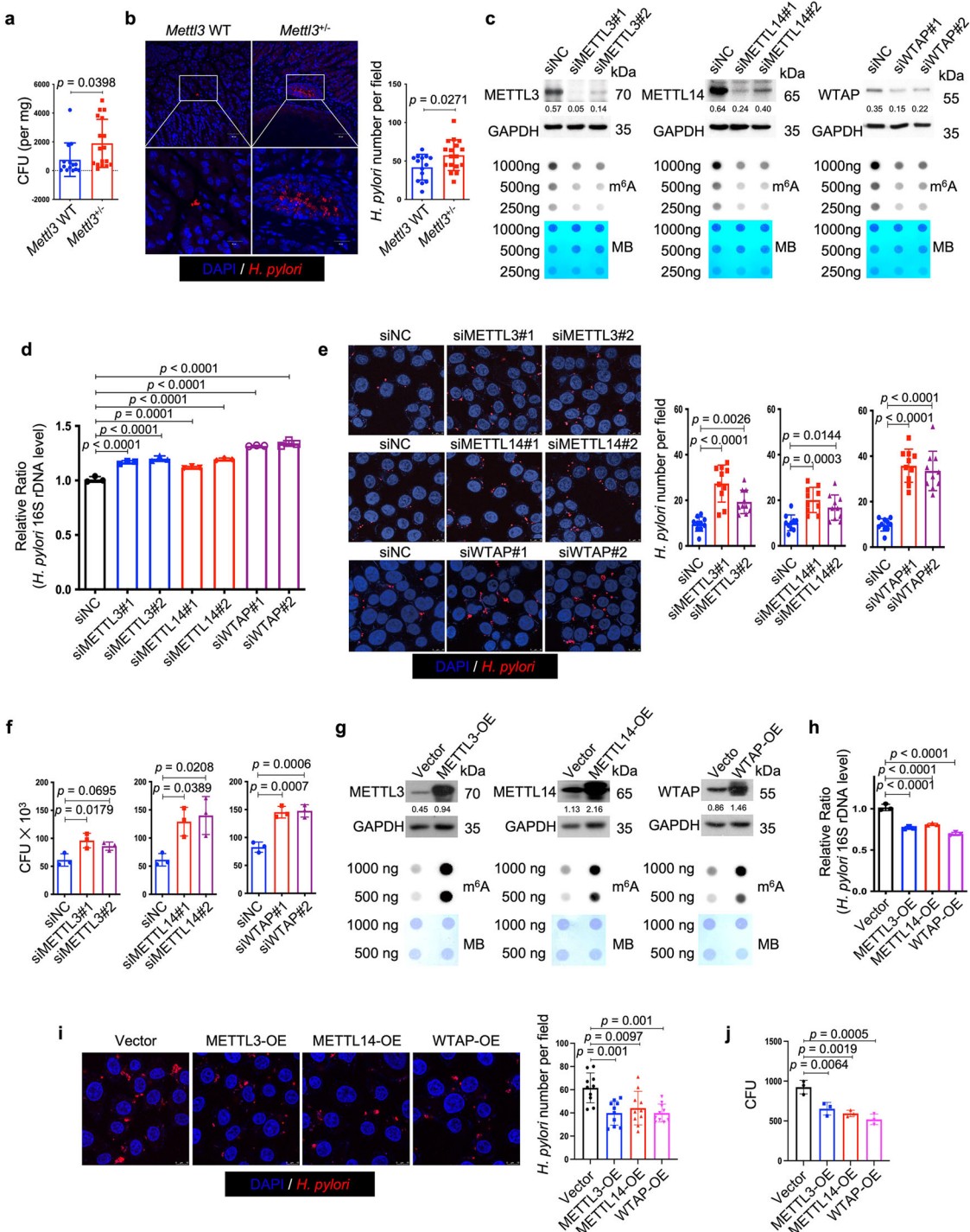

**Fig. 2 | Ablation of m⁶A writers enhanced whereas overexpression of m⁶A writers reduced *H. pylori* infection.** **a**, **b** Wild type (*n* = 14 animals) and Mettl3⁺/⁻ mice (*n* = 17 animals) were infected with *H. pylori* strain SS1 for 3 months. **a** Mouse stomach tissues were harvested and weighted, and then homogenized in sterile PBS. The samples were further diluted and spread on *H. pylori*-selective blood agar plates. After growing for 5 days, the colony numbers were count for quantitative analysis. **b** Parafilm-embedded sections of mouse stomach were stained to visualize *H. pylori* (Red) and the nuclei (Blue). Ten visual fields of each sample were randomly selected to count *H. pylori* number. Scale bar, 50 μm. **c**, **g** Two individual siRNAs or overexpression plasmids were transfected into HFE145 cells to knock down or overexpress METTL3, METTL14 and WTAP, respectively. The knockdown or overexpression efficiency was examined by Western blots. GAPDH served as a loading control. The m⁶A level of poly(A)⁺ RNA was examined by Dot blot. Methylene blue (MB) staining was used as a loading control. **d–f**, **h–j** HFE145 cells transfected with siRNAs or overexpression plasmids were infected with *H. pylori* (MOI = 100) for 24 h. **d**, **h** Intracellular *H. pylori* 16S ribosomal DNA levels were measured by real-time PCR. Human GAPDH was used as an internal control (*n* = 3 replicates for each group). **e**, **i** Cells were stained to visualize invaded *H. pylori* (Red) and the nuclei (Blue). Ten visual fields of each group were randomly selected to count *H. pylori* number. **f**, **j** Cells were permeabilized with 1% saponin for 15 min. The diluted samples were then spread on *H. pylori*-selective blood agar plates and incubated for 5 days to count the colony number (*n* = 3 replicates for each group). All the quantitative data were shown as means ± SD. Statistical analysis of the data was performed using unpaired two-tailed Student's *t* test (**a**, **b**) or one-way ANOVA (**d–f**, **h**, **i**, **j**) followed by Tukey's multiple comparison tests with adjustments, and the corresponding *p*-values are included in the figure panels. The statistical significance of the data (**d–f**, **h**, **i**, **j**) was calculated from one of three independent experiments with similar results. Source data are provided as a Source Data file.

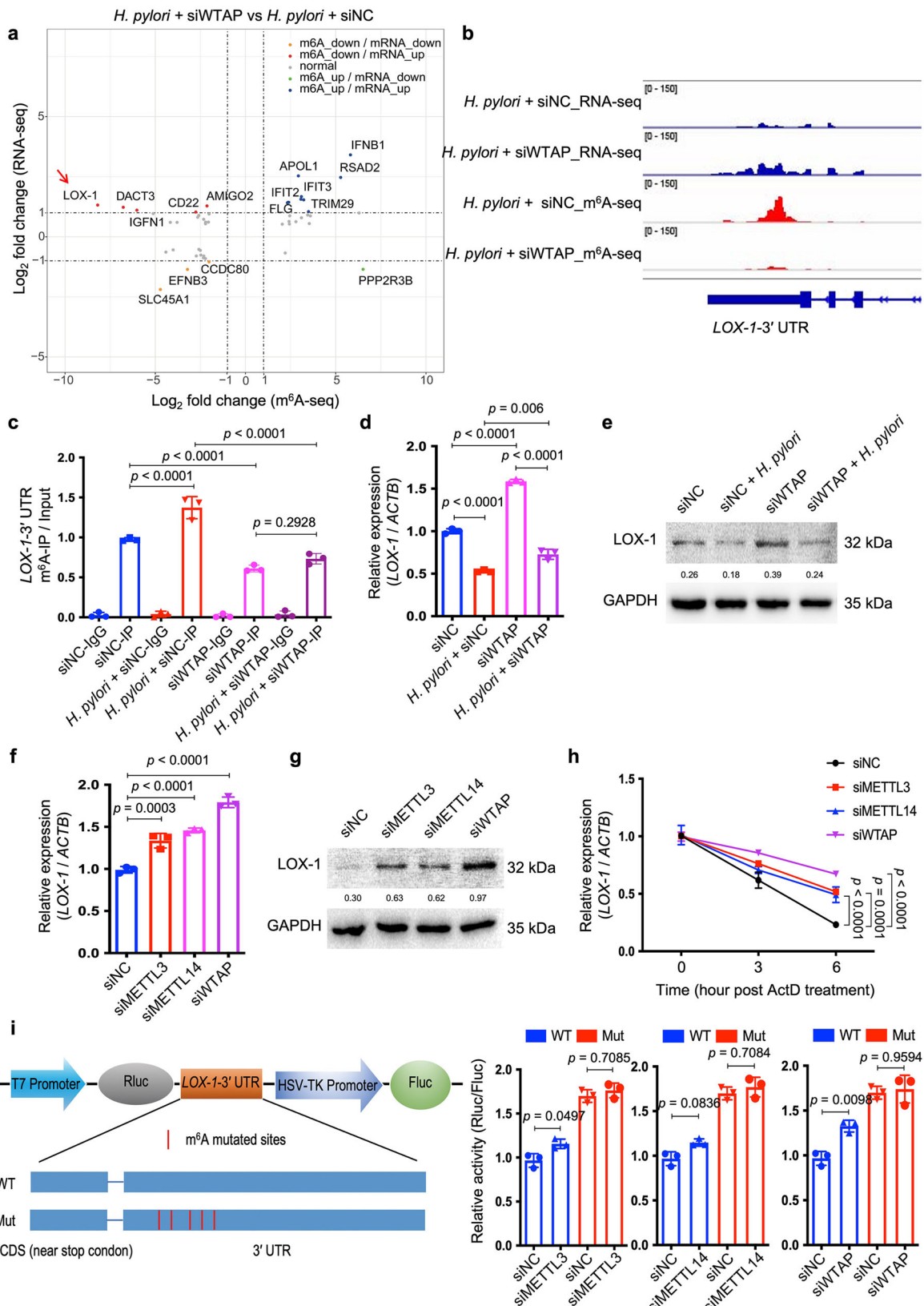

cellular uptake of oxLDL. Combining biophysical and crystal-structure analyses, it was found that BI-0115 binding could inhibit LOX-1 by sta-bilizing an inactive receptor tetramer state[28]. Therefore, we next employed BI-0115 to determine if pharmacological blockage of LOX-1 could suppress *H. pylori* infection. Pre-treating HFE145 cells with 2.5 μM BI-0115 effectively reduced the number of invaded *H. pylori*,

while this effect was not detected when using BI-1580, a control che-mical analog of BI-0115 with much less drug potency (https://www.opnme.com/molecules/lox-1-bi-0115, Fig. 6a–c). Besides, BI-0115 but not BI-1580 significantly inhibited the adhesion of *H. pylori* to HFE145 cells (Fig. 6d). Furthermore, SS1-infected C57BL/6 mice were treated with 10 mg/kg BI-0115 or vehicle on alternate days for 2 weeks followed

**Fig. 3 | LOX-1 was identified as a m⁶A-modified gene upon *H. pylori* infection.**
**a** Integrated analysis of m⁶A-seq and RNA-seq from *H. pylori*-infected HFE145 cells transfected with negative control or WTAP siRNAs was shown. The fold change of m⁶A-seq was normalized to the corresponding transcript expression level. Genes with Log₂ fold change > 1 were highlighted. **b** The m⁶A peaks located at 3′UTR of LOX-1 in m⁶A-seq and the corresponding peaks in RNA-seq were visualized by IGV tool. **c**–**e** HFE145 cells transfected with negative control or WTAP siRNAs were infected with or without *H. pylori* for 24 h. **c** MeRIP-quantitative PCR was performed to validate m⁶A enrichment on LOX-1-3′UTR. The m⁶A enrichment of each group was calculated by m⁶A-IP/input (*n* = 3 replicates for each group). **d** *LOX-1* mRNA levels of each group were measured by real-time PCR. Human *ACTB* was used as an internal control (*n* = 3 replicates for each group). **e** LOX-1 protein levels of each group were examined by Western blots. Human GAPDH was used as a loading control. **f**–**h** HFE145 cells transfected with negative control or siRNAs targeting METTL3/METTL14/WTAP were infected with *H. pylori* for 24 h. **f** *LOX-1* mRNA levels of each group were measured by real-time PCR. Human *ACTB* was used as an

internal control (*n* = 3 replicates for each group). **g** LOX-1 protein levels of each group were examined by Western blots. Human GAPDH was used as a loading control. **h** RNA decay rates of *LOX-1* of each group were measured after treating with Actinomycin D (normalize to 0 h, *n* = 3 replicates for each group). **i** Two luciferase plasmids were constructed by inserting corresponding CDS into pmiR-RB-Report™ vector. The wild type plasmid contained the full-length 3′UTR of LOX-1 and partial CDS near the stop codon, whereas five m⁶A-consensus motifs identified from m⁶A-seq were mutated with A-to-C conversion in the mutated plasmid. The relative luciferase activity was measured and calculated by normalizing Renilla to Firefly activity (*n* = 3 replicates for each group). All the quantitative data were shown as means ± SD. Statistical analysis of the data was performed using one-way ANOVA (**c**, **d**, **f**, **h**, **i**) followed by Tukey's multiple comparison tests with adjustments and the corresponding *p*-values are included in the figure panels. The statistical significance of the data (**c**, **d**, **f**, **h**, **i**) was calculated from one of three independent experiments with similar results. Source data are provided as a Source Data file.

by colony formation assay and immunofluorescence (Fig. 6e). The results showed that BI-0115 significantly suppressed colonization of *H. pylori* in the mouse stomachs (Fig. 6f, g). Besides, histopathological assessment revealed the attenuation of inflammatory cell infiltration (Fig. 6h). Meanwhile, BI-0115 administration reduced the expression of the pro-inflammatory interleukin-6 (IL-6, Fig. 6i), suggesting that *H. pylori*-induced inflammation was attenuated in BI-0115-treated mouse stomachs. Thus, our in vitro and in vivo findings suggest that targeting LOX-1 may help reduce *H. pylori* colonization in the stomach.

## Discussion

In the present study, we uncovered the up-regulation of key m⁶A methyltransferases, namely METTL3, MELL14 and WTAP, and an elevated cellular m⁶A level in human and mouse *H. pylori*-infected gastric epithelium. It has been reported that the localization and activity of m⁶A writers and erasers are dynamically altered during host response to infection. Srinivas et al. revealed that herpes simplex virus (HSV-1) infection orchestrated a striking redistribution of m⁶A methyltransferases, with METTL3 and METTL14 dispersed into the cytoplasm and WTAP remained inside nucleus[29]. Likewise, activation of the type I interferon signaling in peripheral blood mononuclear cells led to the degradation of WTAP via the ubiquitination-proteasome pathway[30]. A recent study reported that an increase in m⁶A methylation in intestinal epithelial cells following *Cryptosporidium parvum* infection was associated with the downregulation of FTO and ALKBH5[31]. It would be of interest to investigate the upstream pathways that contribute to the upregulation of METTL3, MELL14 and WTAP in *H. pylori*-infected gastric epithelium. We next explored whether the alteration of cellular m⁶A modifications could modulate *H. pylori* infection, and found that silencing major m⁶A "writers" could remarkably promote *H. pylori* colonization in vivo and in vitro. When conducting m⁶A-seq and RNA-seq to identify the potential m⁶A-modified target(s) involved in *H. pylori* infection, it is noteworthy that upon WTAP knockdown the m⁶A levels of a couple of genes, including interferon-related IFNB1 and IFIT2/3, were increased. WTAP was generally considered as a regulatory subunit guiding m⁶A "writer" complex to the proper target sites[32]. This phenomenon suggested that the residual WTAP might still exert some functions, or some other unknown components of the m⁶A "writer" complex might compensatorily take its place. Concerning LOX-1, its m⁶A levels were most significantly decreased after WTAP knockdown. A series of molecular experiments confirmed that the m⁶A levels on 3′-UTR of LOX-1 negatively regulated its expression. However, our reporter assay with five m⁶A sites mutated only led to a 50-60% restoration of the reporter signals, indicating that the m⁶A modification on 3′-UTR of LOX-1 plays a partial role in regulating its expression. Thus, we could not exclude the possibility that other regulatory mechanisms could modulate the expression levels of LOX-1 upon *H. pylori* infection, which was worthwhile to further investigate in the future.

Adhesion of *H. pylori* to human gastric epithelial cells is crucial for bacterial colonization. Moreover, it has been reported that bacterial attachment to host cells surface is an initial step for *H. pylori* invasion and may contribute to the chronic infection[33]. Accumulating evidence has revealed the facultative intracellular nature of *H. pylori*. A previous study reported that approximately 1% of *H. pylori* reside inside gastric epithelial cells[22]. Capurro et al. further showed that intraepithelial residence protected *H. pylori* from antibiotic eradication therapy in the murine model[34]. Moreover, Beer et al. found that intracellular staining of *H. pylori* in patient gastric biopsies was a risk factor for the failure of first-line triple therapies[35]. To date, more than 30 outer membrane proteins (OMPs) of *H. pylori* have been found to be involved in the bacterial attachment to gastric epithelial cells[36,37]. The major OMPs essential for *H. pylori* colonization and infection in gastric mucosa mainly include the blood group antigen-binding adhesion (BabA) that binds to host cell surface Lewis b glycan (Leᵇ) and the sialic acid-binding adherence (SabA) that binds to the sialylated Lewis x (sLeˣ) and Lewis a (sLeᵃ)[38]. In this context, anti-adhesive therapy targeting OMP-receptor interactions has been proposed as a novel strategy to prevent or cure *H. pylori* infection[39,40]. However, previous pre-clinical and clinical studies revealed that 3′-sialyllactose sodium salt, an oligosaccharide that exhibited a strong inhibitory effect on *H. pylori* adhesion in vitro, has limited effect on suppressing or curing *H. pylori* infection[41,42], indicating that other unknown host-bacteria interactions might be involved and affect the efficiency of anti-adhesive therapy. In this study, we identified LOX-1 as a novel host cellular membrane receptor, which is responsible for *H. pylori* adhesion and the subsequent invasion into the gastric epithelial cells. As reported, LOX-1 is classified as the class E1 of scavenger receptors (SR-E1). Scavenger receptors (SRs) are a large family of cell surface receptors which was firstly identified in macrophages that contributed to the clearance of modified (acetylated or oxidized) low-density lipoproteins. The subsequent studies showed that SRs not only could recognize damage-associated molecular patterns (DAMPs), like oxLDL, but could also bind to pathogen-associated molecular patterns (PAMPs), such as lipoteichoic acid (LTA) and lipopolysaccharide (LPS) of Gram-positive and Gram-negative bacteria, thus leading to the clearance of these pathogens through various mechanisms, like adhesion, endocytosis and phagocytosis[43,44]. In particular, surface-associated GroEL protein, a 60 kDa heat shock protein (HSP), was found to promote the adhesion of *Escherichia coli* and *Mycobacterium tuberculosis* to macrophages through binding with LOX-1[25,26]. In the present study, we further identified catalase as the *H. pylori* protein that binds to LOX-1 to initiate the invading process. In this regard, it is generally accepted that catalase is a major antioxidant enzyme in organisms by catalyzing hydrogen peroxide to water and oxygen. However, an increasing number of macromolecules were identified as "moonlighting" proteins, of which unrelated functions beside their well-established ones

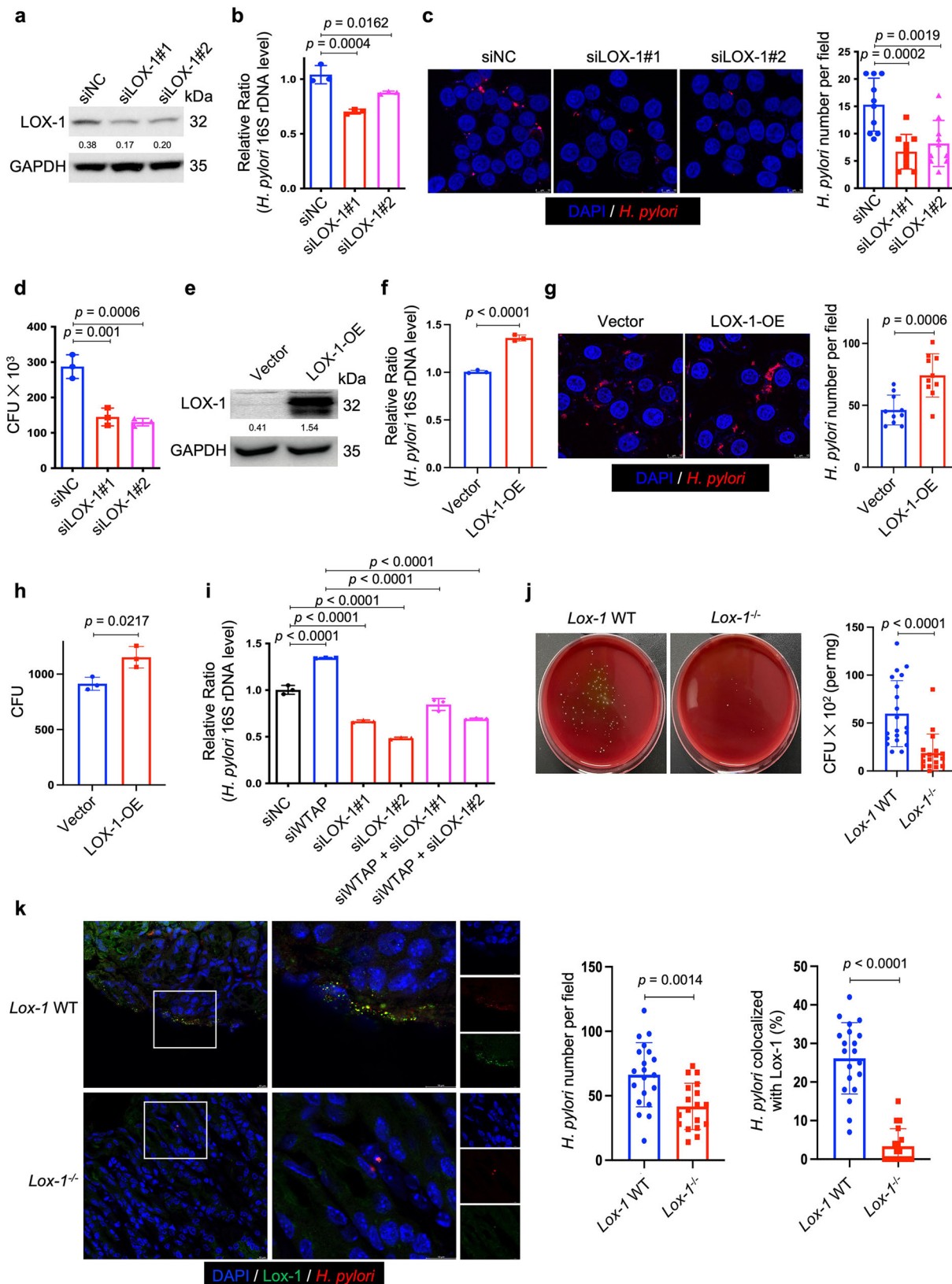

were identified. These proteins, including GroEL, have also been reported to moonlight as bacterial adhesins[45,46]. Based on our current findings, the identified LOX-1-catalase interaction may thus serve as a potential drug target for alleviating *H. pylori* infection. Nevertheless, it is noteworthy that blocking the interaction between LOX-1 and catalase by using LOX-1 blocker can only reduce *H. pylori* adhesion and colonization to some extent but not completely, and catalase knockout also only reduced the adhesion of *H. pylori* to human gastric body and antrum tissues by 48.8% and 42.2%, respectively, and showed modest effect (11.8% inhibition) in cardia tissues, indicating that other bacteria-host interaction could still partially support *H. pylori* colonization. BabA is the most well-described *H. pylori* outer membrane

**Fig. 4 | LOX-1 regulated intracellular *H. pylori* survival in gastric epithelial cells.** **a**, **e** Two individual siRNAs or overexpression plasmids were transfected into HFE145 cells to knock down or overexpress LOX-1. The knockdown or over-expression efficiency was validated by Western blots. GAPDH was served as a loading control. **b**–**d**, **f**–**h** HFE145 cells transfected with siRNAs or overexpression vector of LOX-1 were infected with *H. pylori* (MOI = 100) for 24 h. **b**, **f** Intracellular *H. pylori* 16S ribosome DNA levels were measured by real-time PCR. Human GAPDH was used as an internal control (n = 3 replicates for each group). **c**, **g** Cells were stained to visualize invaded *H. pylori* (Red) and nuclei (Blue). Ten visual fields of each group were randomly selected to count *H. pylori* number. **d**, **h** Cells were permeabilized with 1% saponin for 15 min. The diluted samples were then spread on *H. pylori*-selective blood agar plates and incubated for 5 days to count the colony number (n = 3 replicates for each group). **i** HFE145 cells transfected with WTAP siRNAs and/or LOX-1 siRNAs were infected with *H. pylori* (MOI = 100) for 24 h. Intracellular *H. pylori* 16S ribosomal DNA levels were measured by real-time PCR. Human GAPDH was used as an internal control (n = 3 replicates for each group).

**j**, **k** Wild type (n = 20 animals) and *Lox-1⁻/⁻* mice (n = 18 animals) were infected with *H. pylori* strain SS1 for 1 month. **j** Mouse stomach tissues were harvested and weighted, and then homogenized in sterile PBS. The samples were further diluted and spread on *H. pylori*-selective blood agar plates. After growing for 5 days, the colony numbers were count for quantitative analysis. **k** Parafilm-embedded sections of mouse stomach were stained to visualize Lox-1 (green), *H. pylori* (Red) and the nuclei (Blue). Ten visual fields of each sample were randomly selected to count *H. pylori* number and measure the colocalized signals of *H. pylori* and Lox-1. Scale bar, 10 μm. All the quantitative data were shown as means ± SD. Statistical analysis of the data was performed using unpaired two-tailed Student's *t* test (**f**, **g**, **h**, **j**, **k**) or one-way ANOVA (**b**–**d**, **i**) followed by Tukey's multiple comparison tests with adjustments, and the corresponding *p*-values are included in the figure panels. The statistical significance of the data (**b**–**d**, **f**, **g**–**i**) was calculated from one of three independent experiments with similar results. Source data are provided as a Source Data file.

protein which can interact with host cell Lewis[b] blood group antigens to facilitate bacterial adhesion on gastric surface[47]. Actually, previous studies have already shown that the *H. pylori* strains we used in our in vitro studies, namely TN2GF4 and ATCC 43504, could express functional BabA proteins, but was almost undetectable in 26695 strains[48,49]. These results were confirmed by detecting babA gene expression in our laboratory's *H. pylori* strains using semi-quantitative PCR (Supplementary Fig. 12a). Besides, we confirmed that HFE145 cells could express Lewis[b] antigens, with the human gastric adenocarcinoma cell line AGS that has been reported to express Lewis[b] as a positive control[48], and by knocking down two genes responsible for the biosynthesis of Lewis[b] antigens (i.e., *FUT2* and *FUT3*) as a negative control[50] (Supplementary Fig. 12b–d). Therefore, the involvement of other adhesins and their therapeutic targeting need to be taken into consideration when devising novel anti-adhesion therapies.

In summary, we discovered that LOX-1 is a receptor for *H. pylori*. This surface protein is downregulated during *H. pylori* infection via an m⁶A-dependent manner as a host response to attenuate the bacterial adhesion and invasion. Pharmacological blockade of LOX-1 reduces *H. pylori* colonization. These findings not only highlight the importance of m⁶A modification for host cells to defend against *H. pylori* infection, but also provide evidence that targeting LOX-1 might help to control *H. pylori* infection.

## Methods

### Bacteria and cell culture
The *H. pylori* strains TN2GF4, ATCC 43504, 26695, Sydney strain 1 (SS1) were obtained from the Department of Microbiology, the Chinese University of Hong Kong. The stored (at −80 °C) *H. pylori* strains were spread on the horse blood agar plates (Thermo Fisher, PB0122A) and grown in an anaerobic jar with a microaerobic environment (Oxoid CampyGen 2.5 L Sachet, Thermo Fisher, CN0025A) at 37 °C for 4−5 days. Human gastric epithelial cell line HFE145 was established from the laboratory of Prof. Hassan Ashktorab (Howard University, Washington, D.C., USA) and Dr. Duane T. Smoot (Meharry Medical College, TN, USA)[17,18]. AGS cells were from Prof. Jun Yu's team (The Chinese University of Hong Kong, HKSAR, China). HFE145 and AGS cells were cultured in Dulbecco's Modified Eagle Medium (DMEM, Gibco, Thermo Fisher, 11995065) supplemented with 10% Fetal Bovine Serum (FBS, Gibco, Thermo Fisher, A4766801) at 37 °C with 5% CO₂.

### Human gastric specimens and immunohistochemistry staining
*H. pylori*-negative and -positive human gastric specimens were collected from patients undergone gastroduodenoscopy or gastrectomy at the Digestive Endoscopy Center of the First Affiliated Hospital of Nanchang University for upper gastrointestinal symptoms. All these patients were diagnosed with chronic gastritis by gastroscopy and

pathological examination, and had never received any *Helicobacter pylori* eradication therapy before. All specimens were taken from the gastric antrum. The inclusion and exclusion criteria were listed in Supplementary Table 3. The study was approved by the Research Ethics Board of the First Affiliated Hospital of Nanchang University (CDYFYYLK (01-009)). Human gastric antrum specimens before and after treatment with antibiotics were collected form patients who underwent gastric endoscopy at the Prince of Wales Hospital (Hong Kong Special Administrative Region, China). Collected specimens underwent clinical and histological diagnosis to determine the condition of *H. pylori* infection. All patients gave written informed consent on the use of clinical specimens for research purposes. This study was approved by the Joint CUHK-NTEC Clinical Research Ethics Committee (CRE-2014.133-T). IHC staining was preformed to detect the expression levels of major m⁶A "writers" and cellular m⁶A levels in gastric specimens. Briefly, 10% formaldehyde-fixed tissues were sectioned into 4 μm slides. The slides were firstly deparaffinized and rehydrated. Then, 3% H₂O₂ was used to block endogenous peroxidase activity, and citrate buffer antigen retrieval was performed by microwave oven heating. The tissue sections were then incubated with primary antibodies (anti-METTL3 (Cell Signaling Technology, 96391S, 1:50), anti-METTL14 (Cell Signaling Technology, 51104S, 1:50), anti-WTAP (Cell Signaling Technology, 56501S, 1:50)), at 4 °C overnight, followed by incubation of secondary antibody at room temperature for 1 h. Finally, the staining was developed with diaminobenzidine (DAB) solution. Secondary antibody and DAB solution were contained in commercial IHC test kit (Millipore, DAB150). Staining of m⁶A was performed using IHCeasy m⁶A Ready-To-Use IHC Kit (Proteintech, KHC0143) according to the manufacturer's instruction.

### Mice
To generate the *Mettl3⁺/⁻* mice (C57BL/6 background), two specific single-guide RNAs (sgRNAs) targeting exon 2 of *Mettl3* were used, which resulted in a frameshift mutation and generated a premature stop codon. *Mettl3* wild type and *Mettl3⁺/⁻* mice of 6−8 weeks old were orally inoculated with 10⁹ CFU of *H. pylori* strain SS1 on alternate days for a total of 3 doses. Three months after the inoculation, mice were sacrificed, and the stomachs were collected. To generate the *Lox-1⁻/⁻* mice (C57BL/6 background), two specific single-guide RNAs (sgRNAs) targeting the region including exon 5 and exon 6 of *Lox-1* were used, which resulted in a frameshift mutation and generated a premature stop codon. Mice were housed under 12-hour light/dark cycles in a pathogen-free room with clean bedding and free access to food and water, and temperature and humidity were kept at 22−26 °C, 55% ± 5%. Cage and bedding changes were performed each week. All animal studies were performed in accordance with the guidelines approved by the Animal Experimentation Ethics Committee of The Chinese University of Hong Kong and Nanchang University.

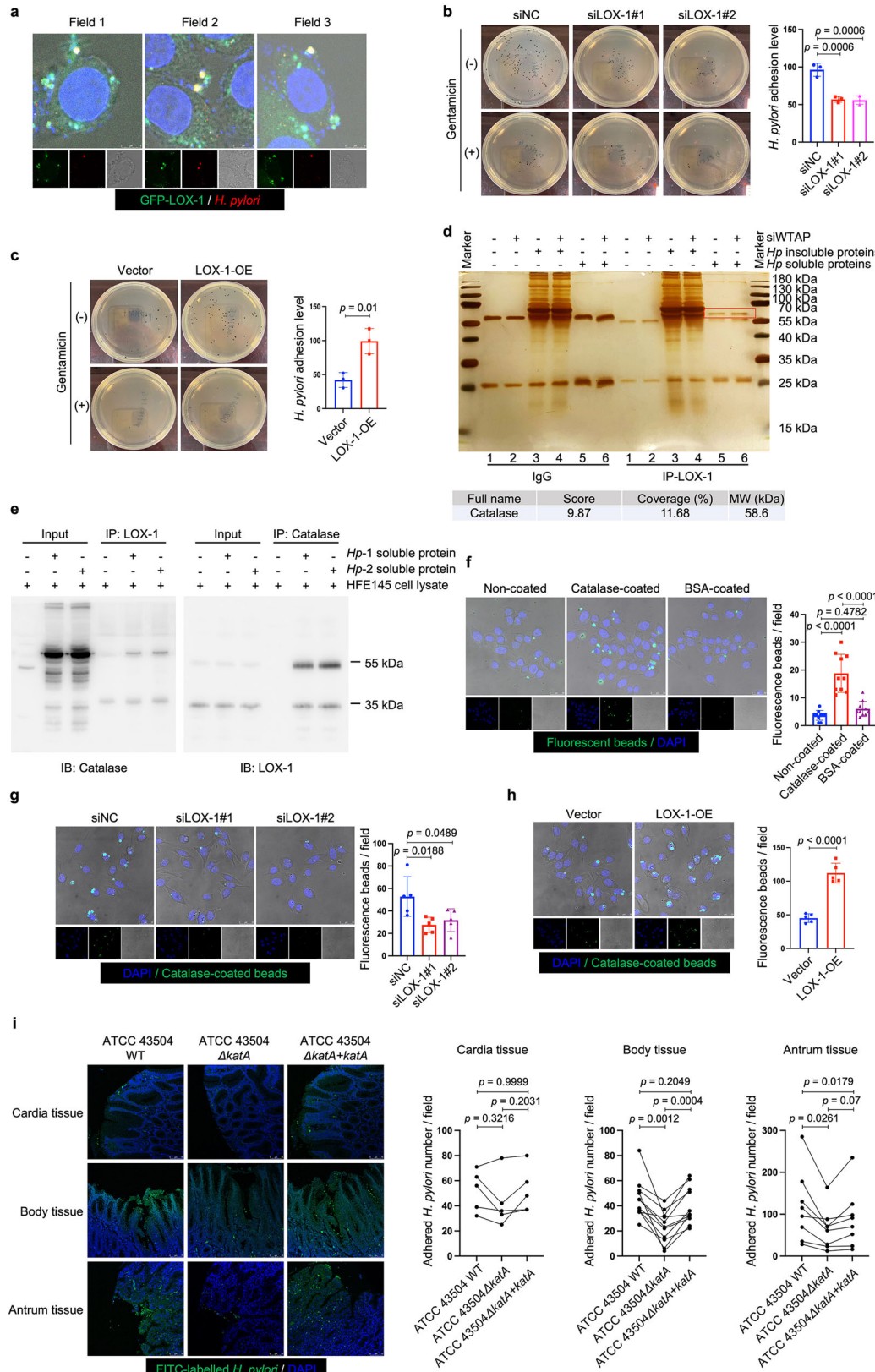

## Murine *H. pylori* infection model and colony-forming assay

*H. pylori* were grown in 200 mL brucella broth (BD, 211088) supplemented with 10% FBS (Gibco, Thermo Fisher, A4766801) for 20–30 h at 37 °C under microaerobic conditions and saturated humidity, with shaking at 200 rpm. C57BL/6 mice were orally inoculated with $10^9$ colony-forming units (CFU) of *H. pylori* strain SS1 or sterile brucella broth on alternate days for a total of 3 doses at 6 weeks of age, and the infection lasted for 3 months. At the end of the study, mice were sacrificed under anesthesia. The mouse stomach was opened along the inner curvature and divided into several parts for different experiments. Colony formation assay was employed to evaluate the *H. pylori* colonization in mouse stomach. In brief, half of the stomach section

**Fig. 5 | LOX-1 was a cell surface protein mediating *H. pylori* adhesion. a** HFE145 cells expressing GFP-LOX-1 (green) protein were infected with *H. pylori* (MOI = 100) for 6 h, and cells were further stained to visualize *H. pylori* (red). The representative visual fields shown are representative of three independent experiments with similar results. Scale bar, 5 μm. **b, c** HFE145 cells transfected with siRNAs or over-expression plasmid of LOX-1 were infected with *H. pylori* (MOI = 100) for 6 h, followed by treating with or without 200 μg/ml gentamycin for 2 h. Cells were permeabilized with 1% saponin for 15 min. The diluted samples were then spread on *H. pylori*-selective blood agar plates and incubated for 5 days to count the colony number. *H. pylori* adhesion levels were calculated by subtracting the CFU of group with gentamycin treatment from the CFU of the group without (*n* = 3 replicates for each group). **d** The lysates of HFE145 cells transfected with negative control or WTAP siRNAs were incubated with soluble or insoluble proteins of *H. pylori*. A specific antibody was used to pull down LOX-1 and its interacting proteins from the incubated protein mixture. The pulled down proteins were separated in SDS-PAGE followed by silver staining. The red square indicates a protein band that was only observed in the LOX-1-pulled down groups incubated with *H. pylori* soluble proteins. Mass spectrum analysis revealed the specific protein band as *H. pylori* catalase. The gel shown is representative of three independent experiments with similar results. **e** HFE145 cell lysate was incubated with the soluble proteins of two *H. pylori* strains (TN2GF4 and ATCC 43504), and LOX-1 or catalase antibody was used to perform reciprocal co-immunoprecipitation assay. The predicted molecular size of catalase is 60 kDa, and the molecular size of LOX-1 is 50 kDa (mature form) and

32 kDa (precursor). The blots shown are representative of three independent experiments with similar results. **f** HFE145 cells were incubated with non-coated, catalase-coated, or BSA-coated fluorescent beads (green) for 6 h (MOI = 100), and cells were stained to visualize nuclei (blue). Ten visual fields of each group were randomly selected to count the attached and internalized fluorescent beads. **g, h** HFE145 cells transfected with siRNAs or overexpression vector of LOX-1 were incubated with catalase-coated fluorescent beads (MOI 100) for 6 h and were stained to visualize nuclei (blue). Ten visual fields of each group were randomly selected to count the attached and internalized fluorescent beads. Scale bar, 25 μm. **i** FITC-labeled *H. pylori* strains (wild-type, *ΔkatA* and *ΔkatA* + *katA* strains) were respectively incubated with human normal stomach tissue array (72 tissue cores from 24 cases (i.e., triplicate sections for each case), US Biomax, BN01011B). Each line represents the binding of the three *H. pylori* strains in an individual patient as measured by calculating the mean number of adhered *H. pylori* on triplicate sections, respectively. Quantitative analysis of the bacteria binding to different gastric areas (cardia tissues from 5 cases, gastric body tissues from 11 cases, and gastric antrum tissues from 8 cases) were performed, respectively. Scale bar, 75 μm. All the quantitative data were shown as means ± SD. Statistical analysis of the data was performed using unpaired two-tailed Student's *t* test (**c, h**) or one-way ANOVA (**b, f, g, i**) followed by Tukey's multiple comparison tests with adjustments, and the corresponding *p*-values are included in the figure panels. The statistical significance of the data (**b–d, f–h**) was calculated from one of three independent experiments with similar results. Source data are provided as a Source Data file.

was homogenized in phosphate buffered saline (PBS) with a grinding bar. Ten-fold serial dilutions were prepared in PBS and diluted aliquots were spread on *H. pylori* selective agar plates (Thermo Fisher, CM0331). After 5–7 days of culture at 37 °C, colonies on the plates were counted and the CFU number per milligram of mouse stomach was calculated. All animal studies were performed in accordance with the guidelines approved by the Animal Experimentation Ethics Committee of The Chinese University of Hong Kong and Nanchang University.

## Quantification of adhesive and invasive *H. pylori*

Human gastric epithelial cells HFE145 were co-cultured with *H. pylori* with an MOI of 1:100 for 6 h, then 200 μg/ml gentamycin (Sigma-Aldrich, 345814-M) was added for 2 h to kill *H. pylori* in the culture medium and on the cell surface. Cells were then washed with PBS for 3 times and replaced with fresh medium. For knockdown or over-expression experiments, small interfering RNAs (siRNAs) or plasmids were transfected into cells for 24 h in advance. For LOX-1 blockage assay, cells were pre-treated with 2.5 μM BI-0115 or the control chemical analog BI-1580 for 6 h in advance.

*H. pylori* 16S *rDNA* PCR, immunofluorescence and colony-forming assays were applied to quantify invasive *H. pylori* abundance. For 16S rDNA PCR, cells were harvested and DNA from cells was extracted using Wizard Genomic DNA Purification Kit (Promega, A1120) according to the manufacturer's instruction. The extracted DNA was examined by quantitative PCR with specific primers of *H. pylori* 16S rDNA, which is listed in Supplementary Table 4. The relative abundance of *H. pylori* was calculated with the comparative threshold cycle (△△ct) method. For immunofluorescence, cells were fixed with 4% paraformaldehyde. After washing 3 times with PBS, cells were permeabilized by 0.1% Triton X-100, followed by blocking the non-specific antigens with 10% goat serum. The intracellular bacteria were subsequently stained by anti-*H. pylori* antibody (Abcam, ab7788, 1:200) at 4 °C overnight, followed by staining with secondary fluorescence antibody (Alexa Fluor 568 goat anti-Rabbit IgG, Invitrogen, A-11011, 1:500). In addition, cell nuclei were stained with 4′,6-diamidino-2-phenylindole. Sections were evaluated using laser scanning confocal microscopy (Leica Microsystems, Wetzlar, Germany). *H. pylori* number was quantitated by random selection of 10 visual fields from each group. For colony-forming assay, cells were lysed with 1% saponin (dissolved in PBS and filtered with 0.22-μm polyvinylidene fluoride filter) for 15 minutes. The lysates were transferred to a clean 1.5 mL tube and a ten-fold serial dilution of samples was prepared.

Serial-diluted lysates were spread on *H. pylori* selective agar plates (Thermo Fisher, CM0331). After 5 days of culture at 37 °C under microaerobic conditions and saturated humidity, colonies were counted and the CFU number was calculated. Colony-forming assay was also applied to quantify adhesive *H. pylori* levels. Similarly, HFE145 cells were first co-cultured with *H. pylori* at an MOI of 1:100 for 6 h, and then treated with or without 200 μg/ml gentamycin for 2 h. Cells were washed with PBS for 3 time and lysed with 1% saponin for 15 min, and the diluted samples were spread on *H. pylori*-selective agar plates. Colonies were counted after 5 days of culture, and adhesion levels were calculated as: $((\text{CFU}_{\text{Gentamycin}-} - \text{CFU}_{\text{Gentamycin}+})_{\text{treatment}} / (\text{CFU}_{\text{Gentamycin}-} - \text{CFU}_{\text{Gentamycin}+})_{\text{control}}) \times 100\%$.

## m⁶A dot blot assay

Total RNA was isolated using RNAiso Plus (Total RNA extraction reagent, Takara, 9109) according to the manufacturers' instructions and quantified by fluorometry. Poly(A)⁺ RNA from total RNA was isolated using the GenElute™ mRNA Miniprep Kit (Sigma, MRN70-1KT) according to the manufacturers' instructions and quantified by Qubit™ RNA HS Assay Kit (Invitrogen, Q32855). The Poly(A)⁺ RNA samples were loaded to the Amersham™ Hybond-N + membrane (GE Healthcare, RPN119B) with a Bio-Dot Apparatus (Bio-Rad, 170-6545) and UV-crosslinked to the membrane. Then the membrane was blocked with 5% non-fat dry milk (in 1 × Tris-buffered saline with 0.1% Tween® 20 detergent) for 1–2 h and incubated with a specific anti-m⁶A antibody (Synaptic Systems, 202003, 1:2000) overnight at 4 °C. Then the membrane was further incubated with horseradish peroxidase (HRP)-conjugated goat anti-rabbit IgG (Cell signaling Technology, 7074S, 1:5000) for 1 h at room temperature. The membrane was developed with Amersham™ ECL Prime Western Blotting Detection Reagent (GE Healthcare, RPN2232). To ensure equal spotting of mRNA onto the membrane, the same blot was stained with 0.02% methylene blue in 0.3 M sodium acetate.

## Immunofluorescence staining

For human gastric epithelial cells, cells were firstly fixed with 4% paraformaldehyde and permeabilized with 0.1% Triton X-100 solution. For mouse stomach sections, slides were firstly deparaffinized and rehydrated, and permeabilized with 0.5% Triton X-100 solution. Then, 10% goat serum was used to block non-specific antigens in both cells and mouse stomach sections. Slides were incubated with primary antibody (anti-*H. pylori* (Abcam, ab7788, 1:200), anti-*H. pylori* (Abcam, ab231433,

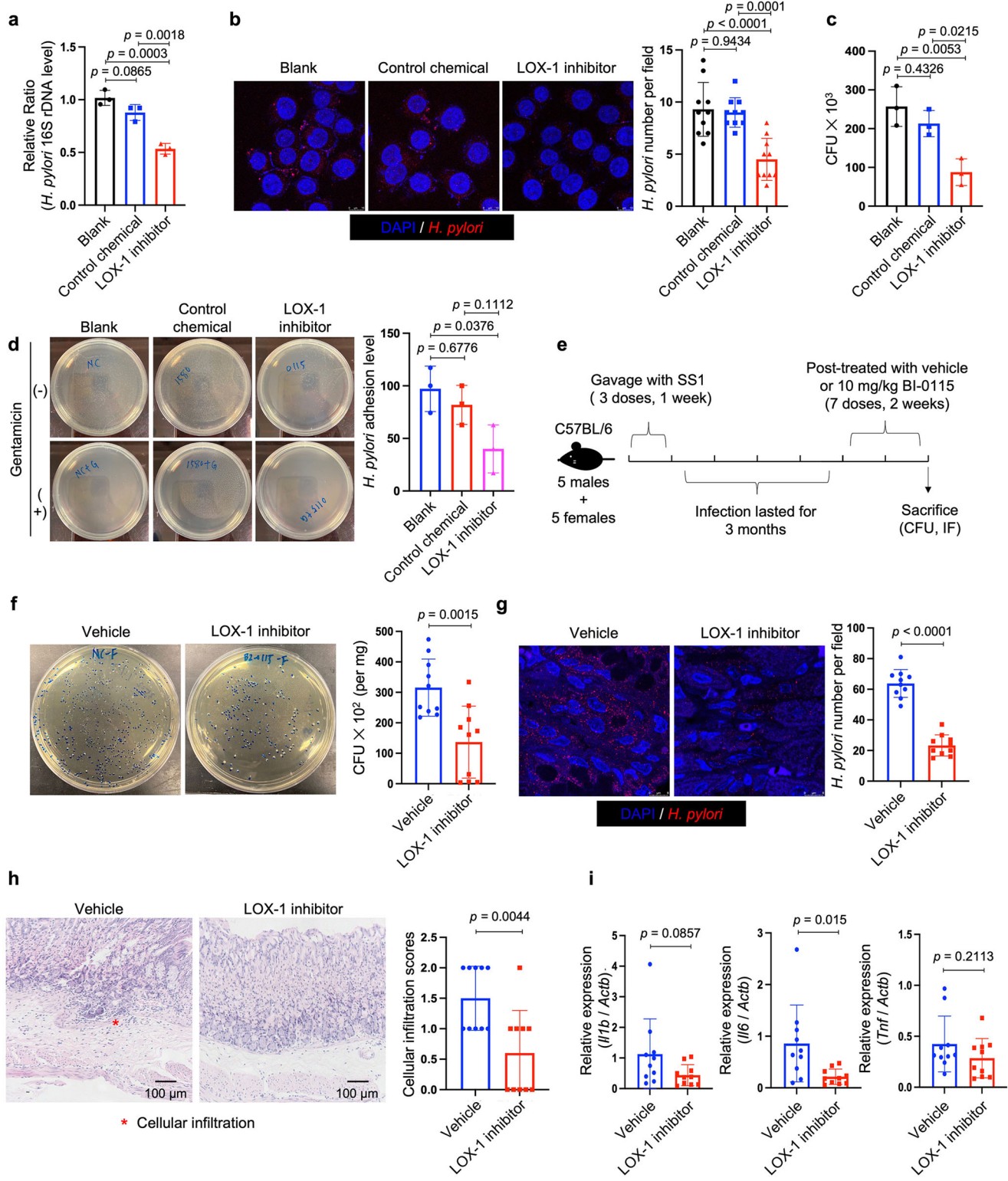

1:200), anti-LOX-1 (Abcam, ab60178, 1:100), anti-His-Tag (Santa Cruz Biotechnology, sc-8036, 1:50), anti-Lewis b (Santa Cruz Biotechnology, sc-51513, 1:50)) 4 °C overnight followed by incubation of secondary antibody (Goat anti-rabbit conjugated to Alexa Fluor 488 (Invitrogen, A-11008, 1:500), goat anti-rabbit conjugated to Alexa Fluor 568 (Invitrogen, A-11011, 1:500), goat anti-mouse conjugated to Alexa Fluor 568 (Invitrogen, A-11031, 1:500)) and then 4′,6-diamidino-2-phenylindole (DAPI) to stain the nucleus. Sections were evaluated using laser scanning confocal microscopy (Leica Microsystems, Wetzlar, Germany).

## Western blots

Tissues or cells were lysed in radioimmunoprecipitation assay (RIPA) buffer. After sonication for 30 seconds on ice and centrifuging for 30 minutes at $15,000 \times g$ at 4 °C, the supernatant was collected, and protein concentration was measured by BCA Protein Assay Kit (Thermo Scientific, 23225). Twenty micrograms of protein of each sample were separated by 6–15% sodium dodecyl sulfate polyacrylamide gel electrophoresis (SDS-PAGE). Proteins on the gels were then transferred onto a polyvinylidene Fluoride (PVDF) membrane.

**Fig. 6 | Administration of BI-0115, a LOX-1 inhibitor, suppressed *H. pylori* infection in vitro and in vivo. a**–**d** HFE145 cells pre-treated with LOX-1 inhibitor (BI-0115, 2.5 µM) or control chemical analog (BI-1580, 2.5 µM) were infected with *H. pylori* (MOI = 100) for 6 h. **a** Intracellular *H. pylori* 16 S ribosomal DNA levels were measured by real-time PCR. Human GAPDH was used as an internal control (*n* = 3 replicates for each group). **b** Cells were stained to visualize invaded *H. pylori* (Red) and nuclei (Blue). Ten visual fields of each group were randomly selected to count the *H. pylori* number. Scale bar, 10 µm. **c** Cells were permeabilized with 1% saponin for 15 min. The diluted samples were then spread on *H. pylori*-selective blood agar plates and incubated for 5 days to count colony number (*n* = 3 replicates for each group). **d** Cells were treated with or without 200 µg/ml gentamycin for 2 h, and permeabilized with 1% saponin for 15 min. The diluted samples were then spread on *H. pylori*-selective blood agar plates and incubated for 5 days to count the colony number. *H. pylori* adhesion levels were calculated by subtracting the CFU of the group with gentamycin treatment from the CFU of the group without (*n* = 3 replicates for each group). **e** C57BL/6J mice were orally inoculated with *H. pylori* strain SS1 for 3 months, followed by administration of the vehicle (*n* = 10 animals) or the LOX-1 inhibitor (BI-0115; 10 mg/kg, *n* = 10 animals) on alternate days for 2 weeks.

**f** Mouse stomach tissues were harvested and weighted, and then homogenized in sterile PBS. The samples were further diluted and spread on *H. pylori*-selective blood agar plates. After growing for 5 days, the colony numbers were count for quantitative analysis (*n* = 10 animals for each group). **g** Parafilm-embedded sections of mouse stomach were stained to visualize colonized *H. pylori* (Red) and the nuclei (Blue). Ten visual fields of each sample were randomly selected to count *H. pylori* number. Scale bar, 25 µm. **h** Parafilm-embedded sections were stained with hematoxylin and eosin (H&E). Histopathological assessment (cellular infiltration: 0–3) was conducted in two separate tissue sections for each animal (*n* = 10 animals for each group). Scale bar, 100 µm. **i** mRNA expressions of pro-inflammatory cytokines (*Il1b*, *Il6*, *Tnf*) in mouse stomach were measured by real-time PCR (*n* = 10 animals for each group). All the quantitative data were shown as means ± SD. Statistical analysis of the data was performed using unpaired two-tailed Student's *t* test (**f**–**i**) or one-way ANOVA (**a**–**d**) followed by Tukey's multiple comparison tests with adjustments, and the corresponding *p*-values are included in the figure panels. The statistical significance of the data (**a**, **c**, **d**, **f**, **h**, **i**) was calculated from one of three independent experiments with similar results. Source data are provided as a Source Data file.

The non-specific binding sites on membrane were blocked with 5% non-fat dry milk in 1 × Tris-buffered saline with 0.1% Tween20 (TBST) for 1 h, and then incubated with the corresponding primary antibody (anti-METTL3 (Cell Signaling Technology, 96391S, 1:1000), anti-METTL14 (Cell Signaling Technology, 51104S, 1:1000), anti-WTAP (Cell Signaling Technology, 56501S, 1:1000), anti-FTO (Abcam, ab124892, 1:1000), anti-ALKBH5 (Abcam, ab69325, 1:1000), anti-LOX-1 (Santa Cruz Biotechnology, sc-66155, 1:200), anti-CagA (Santa Cruz Biotechnology, sc-28368, 1:200), anti-His-Tag (Santa Cruz Biotechnology, sc-8036, 1:200), anti-catalase (constructed by Abclonal company, 1:1000), anti-Lewis b (Santa Cruz Biotechnology, sc-51513, 1:200), anti-ACTB (Cell Signaling Technology, 4967S, 1:2000), anti-GAPDH (Santa Cruz Biotechnology, sc-365062, 1:200)) overnight at 4 °C. After washing with 1 × TBST for 10 min for 3 times, the membranes were further incubated with secondary antibody (anti-rabbit conjugated to horseradish peroxidase (Cell Signaling Technology, 7074S, 1:5000) and anti-mouse conjugated to horseradish peroxidase (Cell Signaling Technology, 7076S, 1:5000)) at room temperature for 1 h. Chemiluminescent signals were then developed with LumiGLO reagent (Cell Signaling Technology, 7003S) and exposed on X-ray film in dark room or using ChemiDoc™ MP Imaging System (Bio-Rad).

## Quantitative reverse transcription PCR

Total RNA was isolated using RNAiso Plus (Total RNA extraction reagent, Takara, 9109) according to the manufacturers' instructions and quantified by NanoDrop 2000 Spectrophotometer. One microgram of total RNA was reverse-transcribed to cDNA using PrimeScript RT Master Mix (Takara, RR036B). Twenty nanograms of DNA or cDNA templates were amplified using TB Green qPCR Premix (Takara, 639676) by QuantStudio™ 12K Flex Real-time PCR system (Thermo Fisher Scientific). The relative expression levels of target genes were calculated with the △△ct method. All the primers sequences we used are listed in Supplementary Table 4.

## RNA interference and gene overexpression

siRNAs targeting METTL3, METTL4, WTAP, LOX-1 and negative control RNAs (siNC) were synthesized by GenePharma Company (Shanghai, China) and Ribo Company (Suzhou, China). 3 × HA-METTL3, METTL14, WTAP, Flag-LOX-1, GFP-LOX-1 plasmids were purchased from Public Protein/Plasmid Library (Jiangsu, China). Transient transfection was performed using jetPRIME Polyplus kit (Polyplus transfection, 114-15) according to the manufacturers' instructions. Cells were harvested after 24–72 h for quantitative RT-PCR examination, Western blot analysis and functional assays. All the siRNA sequences we used are listed in Supplementary Table 5.

## m⁶A-seq and data analysis

Polyadenylated RNA was extracted from treated cells using FastTrack MAG Maxi mRNA isolation kit (Life technology, USA). RNA fragmentation Reagents (Ambion, USA) was used to randomly fragment RNA. The specific anti-m⁶A antibody (Synaptic Systems, 202003, 1:50) was applied for m⁶A pull down. The library preparation for next-generation sequencing was constructed by TruSeq Stranded mRNA Sample Prep Kit (Illumina, USA) and quantified by BioAnalyzer High Sensitivity DNA chip, and then deeply sequenced on the Illumina HiSeq 2500. For data analysis, the reads from input and m⁶A-IP sequencing libraries were aligned to hg19 reference genome using Tophat. Both MACS/MACS2 and exomePeak were used to call m⁶A peaks based on the m⁶A-seq bam files. To achieve high specificity, only the m⁶A peaks called by both MACS/MACS2 and exomePeak were retained for further analysis. Differentially methylated m⁶A peaks were identified according to the procedure described by Schwartz et al.[51]. Sequence motifs enriched in m⁶A peak regions compared to control regions were identified using DREME.

## RNA-seq

Total RNA was isolated using RNAiso Plus (Total RNA extraction reagent, Takara, 9109) according to the manufacturer's instructions and quantified by the NanoDrop 2000 spectrophotometer. TruSeq Stranded Total Sample Preparation Kit (Illumina, 20020596) was used for sequencing library construction. Libraries were deeply sequenced on the Illumina Novaseq 6000 according to the activity and expected data volume. The raw RNA-Seq sequence reads were trimmed using Trimmomatic (version 0.39) to remove low-quality reads and adapters. Trimmed data were first evaluated by the software "FastQC" (https://www.bioinformatics.babraham.ac.uk/projects/fastqc/) with the default parameter and then aligned with Hisat2 (version 2.1.0) against the human (hg38) genome guided by GENCODE gene annotation (version 34) with the default parameter. The abundance of genes in each sample was calculated by StringTie packages (version 2.1.2) with the "-e" parameter. Differentially expressed genes were identified using the R package DESeq2 (version 1.34.0) with the following condition: adjusted *p*-value < 0.05 and the absolute value of $\log_2$ fold-change > 1.

## Methylated RNA immunoprecipitation (MeRIP)-quantitative PCR

Total RNA was extracted form treated cells using RNAiso reagent (Takara, 9109) and then incubated with DNase I (Roche, 04716728001) to remove DNA contamination. Total RNA (200 µg) was randomly fragmented into ~200 nucleotides by RNA Fragmentation Buffer (100 mM Tris-HCl, 100 mM ZnCl₂ in DEPC-treated water). Protein A

magnetic beads (Thermo Fisher, 10002D) were pre-incubated with 5 μg of anti-m⁶A polyclonal antibody (Synaptic Systems, 202003, 1:50) or rabbit IgG isotype control (Cell Signaling Technology, 3900, 1:50), and then incubated with fragmented RNA in IP buffer (150 nM NaCl, 10 mM Tris-HCl, 0.1% IGEPAL CA-630 in DEPC-treated water). The fragmented RNA binding with Protein A beads-antibody mixture was isolated using RNeasy Mini Kit (QIAGEN, 74106) according to the manufacturer's instruction. The eluted RNA was reverse-transcribed to cDNA using High-Capacity cDNA Reverse Transcription Kit (Thermo Fisher, 4368814). The m⁶A levels of interested genes were amplified using specifically designed primers (Supplementary Table 4), which were based on the distribution of m⁶A peaks on the target genes from m⁶A-seq. The m⁶A levels of target genes were calculated as the following formula: $2^{(\text{Ct value IP} - \text{Ct value Input})}$.

### RNA-decay assay
Actinomycin D (Sigma-Aldrich, A1410) was added into cells at a final concentration of 5 μg/mL, and cells were collected after 0, 3, 6 h, respectively. Total RNA was isolated using RNAiso Plus (Total RNA extraction reagent, Takara, 9109) and subject to quantitative RT-PCR to compare the degradation rate of LOX-1 mRNA in each group.

### Luciferase reporter assay
Full-length 3′-UTR of LOX-1 and partial CDS near stop codon was cloned into pmiR-RB-Report™ vector which was obtained from Ribo Company (Suzhou, China). For mutant plasmid, 5 adenosines (A) in m⁶A motif were replaced by cytosines (C). The mutated sites and methods were shown in Supplementary Data 1. Pre-treated cells were transfected with wild type or mutated plasmid using jetPRIME polyplus kit. After 24 h, cells were harvested using Dual-Glo Luciferase Assay System (Promega, E2940). The relative luciferase activity was calculated by normalizing fluorescent value of Renilla to Firefly.

### Lysis of *H. pylori* and proteins extraction
*H. pylori* in culture medium was harvested by centrifugation at $1000 \times g$ for 15 min at 4 °C and the supernatant was discarded. The wet pellet was weighted by using total weight to minus the weight of centrifuge tubes. Approximately 3 mL of lysis buffer (50 mM Tris-HCl, pH 8.0, 0.1 mM EDTA, 50 mM NaCl, 1 mM PMSF) was added for each wet gram of *H. pylori* pellet for resuspension. Lysozyme was added to a concentration of 300 μg/mL and the suspension was stirred for 30 min at 4 °C. Triton X-100 was then added to a concentration of 1% (v/v) and ultrasound sonication was applied for three bursts of 30 s followed by cooling. The suspension was then placed at room temperate with DNase I added to a concentration of 10 mg/mL and MgCl₂ to 10 mM. The suspension was stirred for further 15 minutes to remove the viscous nucleic acid. The suspension was centrifuged at $10,000 \times g$ for 15 min at 4 °C, and the soluble proteins in supernatant were collected and transferred to a clean 1.5 mL EP tube. The insoluble proteins were also collected by resuspending the pellet with the same volume of lysis buffer. The soluble and insoluble proteins were directly used for following study or store at −80 °C freezer.

### Co-incubation of *H. pylori* proteins with lysates of HFE145 cells and co-immunoprecipitation
The lysates of HFE145 cells were extracted with NP-40 lysis buffer (50 mM Tris, pH 7.4, 250 mM NaCl, 5 mM EDTA, 50 mM NaF, 1 mM Na3VO4, 1% NP40, 0.02% NaN3) and then incubated with extracted soluble or insoluble proteins of *H. pylori*, as well as LOX-1-specific antibody (Santa Cruz Biotechnology, sc-66155, 1:50) / catalase-specific antibody (constructed by Abclonal company, 1:50) and protein A/G magnetic beads (Thermo Scientific, 88803) at 4 °C overnight. The magnetic beads were collected by placing the tubes on a magnetic

rack. The proteins bound on the beads were eluted by heating in 50 μL of 2 × SDS loading buffer at 100 °C. The eluted proteins were then separated by SDS-PAGE, and the gel was used for Western blot or silver staining, the latter of which was performed following the instructions of the Silver Stain for Mass Spectrometry kit (Thermo Scientific, 24600). The specific bands detected in silver staining were cut off and sent for mass spectrometry.

### Purification of His-tagged catalase and coating with fluorescence beads
The plasmid pMRLB1 containing gene *katA* (coding for catalase of *H. pylori*) was transformed into *E. coli* BL21 and catalase protein expression was induced using 1 mM isopropyl β- d-1-thiogalactopyranoside (IPTG). Histidine (His) tagged-catalase was purified from the bacterial lysate by Ni-NTA affinity chromatography and dialyzed with PBS. Purity of proteins was analyzed using 10% SDS-PAGE with anti-His antibody. Purified catalase proteins or bovin serum albumin (BSA) (0.5 mg/ml) were then incubated with carboxylate modified fluorescent latex beads (Sigma-Aldrich, L4655) for 2 h on a rotator at room temperature. The beads were wash with PBS to remove unbound proteins. The coating efficiency on latex beads was further confirmed by immuno-fluorescence with anti-His antibody (Santa Cruz Biotechnology, sc-8036, 1:50). HFE145 cells were incubated with catalase- or BSA-coated beads at a ratio of 1:100, and the adhesion and internalization of latex beads were evaluated with confocal microscopy.

### Construction of *H. pylori ΔkatA* mutant strain
The isogenic catalase-negative mutants (*ΔkatA*) were constructed with *H. pylori* strain ATCC 43504 (NCTC 11637) using an allelic exchange method as previously described[52]. In detail, 500 bp of up- and downstream-regions of *katA* gene were PCR-amplified from the extracted genomic DNA template of *H. pylori* strain ATCC 43504 with primer pairs Up_F/Up_R and Down_F/Down_R (Supplementary Table 4), respectively. The kanamycin resistance cassette (*aphA-3*) was also PCR-amplified from the pET-28a plasmid with primer pairs KanR_F / KanR_R (Supplementary Table 4). The 5′ regions of primers Up_R and Down_F contained -20 bp complementary fragments to the 3′ regions of KanR_F and KanR_R, respectively. The three PCR products, namely Up_*katA*, Down_*katA* and KanR, were then subject to electrophoresis and purified using an agarose gel DNA extraction kit (Takara, 9762), and were assembled by two sequential overlap extension PCR reactions. In brief, a template mixture containing 100 ng of Up_*katA* and KanR was first PCR-amplified using primers Up_F and KanR_R to generate a linear construct containing Up_*katA* and KanR, which was further purified and mixed in equal volume with 100 ng of Down_*katA* for PCR-amplification with primers Up_F and Down_R to generate a linear construct containing Up_*katA*, KanR and Down_*katA*. The PCR procedures used were as follows: 35 cycles of 98 °C for 10 s, 59 °C for 5 s and 72 °C for 30 to 60 s/kb using PrimeSTAR Max DNA Polymerase (Takara, R045A). The resulting allelic replacement construct was further purified from agarose gel and was confirmed by sequencing. Next, allelic replacement construct was introduced into ATCC 43504 by natural transformation. Briefly, ATCC 43504 was collected from 72 hour-cultured horse blood agar plate (Thermo Scientific, PP2001) and was suspended in 2 mL Brucella broth. Next, 200 ng purified allelic replacement construct was added into an aliquot of 0.2 mL suspended bacteria culture. The mixture was then spread onto horse blood agar and incubated at 37 °C, 5% CO₂ in a humid chamber for 24 h. The growing bacteria on horse blood agar was then collected and suspended in Brucella broth, and an aliquot of 100 μl bacteria was transferred to the agar plates containing 50 μg/ml kanamycin and cultured for another 5–7 days for resistance selection. The growing single colonies were picked for expanded culture and further sent for sequencing to confirm that the central region of *katA* gene was replaced by the KanR cassette.

### Genetic complementation of *katA*

The primers used for ATCC 43504 (NCTC 11637) *katA* gene amplification (Supplementary Table 4) was designed based on the genomic sequencing data of ATCC 43504 from NCBI GenBank (LS483488.1) with the restriction sites of BamHI (G^GATCC) and NheI (G^CTAGC). *H. pylori* shuttle vector pHel2 (addgene, 102960) was used for genetic complementation[53]. Briefly, pHel2 plasmid was linearized with restriction enzyme BamHI-HF® (NEW ENGLAND BioLabs, R3136S) and NheI-HF® (NEW ENGLAND BioLabs, R3131S). Both PCR-amplified *katA* fragment and linearized pHel2 were purified from agarose gel and ligated using In-Fusion® Snap Assembly Master Mix (Takara, 638948) according to the manufacturer's instructions. The insertion of *katA* fragment into pHel2 was assessed by agarose gel electrophoresis after BamHI-HF® and NheI-HF® restriction enzymes digestion. PCR-amplified *katA* fragment and linearized pHel2 were used as control. Recombinant pHel2 plasmid containing the *katA* fragment (pHel2::*katA*) was electroporated into Δ*katA* mutant according to the published literature using a Bio-Rad gene pulser (2.5 kV, 0.2 cm gap, 25 μF capacity, 200 Ω)[54]. After the pulse, the bacteria were spread onto horse blood agar and incubated at 37 °C, 5% $CO_2$ in a humid chamber for 24 h. The growing bacteria on horse blood agar was then collected and suspended in Brucella broth, and an aliquot of 100 μl bacteria was transferred to the agar plates containing 25 μg/ml chloramphenicol and cultured for another 5–7 days for resistance selection. The expression level of *katA* was determined by Western blots.

### Labeling of *H. pylori* strains and measurement of adhesion to the human gastric tissue sections

An aliquot of 500 μl of a suspension of *H. pylori* in PBS ($OD_{550} = 1.0$) was mixed with 1 μl of FITC (10 mg/ml, Invitrogen, F1906) and incubated in the dark for 1 h. Bacteria were washed three times with PBS (0.05% Tween 20, 1% BSA) and resuspended in 500 μl of PBS for direct use. The human normal gastric tissue arrays (72 tissue cores from 24 cases (i.e., triplicate sections for each case), US Biomax, BN01011B) were baked for 2 hours at 60 °C before use to prevent tissue detachment. The tissue arrays were then deparaffinized by treating the cover slides in xylene (2 × 10 min), 100% ethanol (1 × 5 min), 90% ethanol (1 × 5 min), 80% ethanol (1 × 5 min), 70% ethanol (1 × 5 min), 50% ethanol (1 × 5 min) and ddH₂O (1 × 2 min). After coating the sections with 10% solution of goat serum for blocking, the FITC-labeled *H. pylori* ($10^8$ CFU per ml) were added to the slides and incubated for 1 hour at 37 °C, 5% $CO_2$ in a humid chamber. The cover slides were rinsed three times with PBS to remove unbound bacteria followed by incubation of 4′,6-diamidino-2-phenylindole (DAPI) for 5 min to stain the nucleus and sealed with a cover glass for fluorescence microscopy. The binding of FITC-labeled *H. pylori* to each case was measured by calculating the mean number of bacterial counts on triplicate sections. The % inhibition was calculated as ((Bacterial count$_{WT}$ − Bacterial count$_{\Delta katA}$)/Bacterial count$_{WT}$) per case. The % restoration was calculated as ((Bacterial count$_{\Delta katA+katA}$ − Bacterial count$_{\Delta katA}$)/(Bacterial count$_{WT}$ − Bacterial count$_{\Delta katA}$)) per case.

### Administration of LOX-1 inhibitor

A small amount of LOX-1 selective small molecule inhibitor (BI-0115) and its control chemical analog (BI-1580) used in the in vitro study were ordered for free from "opnMe" organization (https://www.opnme.com/molecules/lox-1-bi-0115, Germany). A larger amount of BI-0115 used in the murine model was purchased from a commercial source (BIOSYNTH Carbosynth, BB178547). For in vitro study, HFE145 cells ($1 \times 10^5$) were seeded in 6-well plates and pre-treated with 2.5 μM BI-0115 (selective small molecule inhibitor of LOX-1) or BI-1580 (control chemical analog of BI-0115) for 6 hours before *H. pylori* infection. Cells were then harvested to quantify the adhesion and invasion of *H. pylori*. For in vivo study, C57BL/6J mice were orally inoculated with $10^9$ CFU of *H. pylori* strain SS1 on alternate days for a total of 3 doses at 6 weeks of age, and the infection lasted for 3 months. Then the mice were given 10 mg/kg BI-0115 (dissolved in 1% DMSO in corn oil) or the equal volume of vehicle of 7 doses on alternate days for 2 weeks. At the end of the study, the mice were sacrificed, and the stomachs were collected for measuring *H. pylori* colonization and histopathological assessment. All animal studies were performed in accordance with the guidelines approved by the Animal Experimentation Ethics Committee of The Chinese University of Hong Kong and Nanchang University.

### Histopathological assessment

Sections of mouse stomachs were evaluated and graded according to the updated Sydney system[55]. For assessment of *H. pylori*-induced gastric inflammation, sections were stained with hematoxylin and eosin (H&E) and scored by two experienced pathologists who were blinded to the experiments. Gastric inflammation was assessed in two separate tissue sections for each animal using the following scoring system: cellular infiltration (migration of lymphocytes and neutrophils into the lamina propria and cellular infiltration below the muscularis mucosae) was graded from 0 to 3, where 0, none; 1, mild widespread; 2, moderate widespread; 3, severe widespread.

### Statistical analysis

All statistical analysis were performed using GraphPad Prism 9.0 software and all quantitative data were shown as means ± standard deviation (SD). Difference between two independent samples were compared by unpaired two-tailed Student's *t* test. Two-tailed paired *t*-test was used to measure the difference of the paired samples. Multiple group comparisons were made by one-way Analysis of Variance (ANOVA) followed by the Tukey's *t* test with adjustments. Pearson's chi-squared test was used to test the difference of categorical data. *P*-values less than 0.05 were considered statistically significant.

### Reporting summary

Further information on research design is available in the Nature Portfolio Reporting Summary linked to this article.

## Data availability

The m⁶A-seq and RNA-seq data generated in this study have been deposited in the Gene Expression Omnibus (GEO) database under accession code GSE220810. Source data are provided with this paper.

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

## Acknowledgements

This project was supported by General Research Fund (14114220 and 14116722 to W.K.K.W.) and Research Matching Grant (8601227 to W.K.K.W.) by the Hong Kong Research Grants Council; Health and Medical Research Fund (19180162 to L.Z.; 21200822 to W.K.K.W.) by the Health Bureau of Hong Kong SAR Government; National Natural Science Foundation of China (81873560 to L.Z.; 81974070 to W.G.; 82100599 and 81960112 to C.X.; 82070576 to W.K.K.W.) by the Ministry of Science and Technology of China, Guangdong Basic and Applied Basic Research Foundation (2023A1515011685 to W.G. and 2023A1515030071 to W.H.), Shenzhen Science and Technology Program (JCYJ20180307150626228 to L.Z.; JCYJ20210324131010027 to W.G.; JCYJ20220530154205011 and JCYJ20230807142314030 to W.H.; JCYJ20180508161604382 to W.K.K.W.) by the Shenzhen Science and Technology Innovation Commission; and The TUYF Charitable Trust (to W.K.K.W. and W.H.).

## Author contributions

W.K.K.W., W.H., M.T.V.C. and L.Z. designed the study, supervised the study progress, and revised the manuscript. J.D.Z. and C.X. conducted the major experiments and prepared the manuscript. Z.H.H. performed the bioinformatic analysis. Q.L. and H.C. helped to conduct the animal experiments. C.H.C., H.A., D.T.S., S.H.W., J.Y., W.G., C.L., H.X., H.R.C., X.D.L., J.C.Y.W., M.I., T.G., L.Z. offered the technical and material support.

## Competing interests

The authors declare no competing interests.

## Additional information

[1]State Key Laboratory of Digestive Diseases, Li Ka Shing Institute of Health Sciences, The Chinese University of Hong Kong, Hong Kong Special Administrative Region, Hong Kong, China. [2]Department of Anaesthesia and Intensive Care and Peter Hung Pain Research Institute, The Chinese University of Hong Kong, Hong Kong Special Administrative Region, Hong Kong, China. [3]CUHK Shenzhen Research Institute, Shenzhen, China. [4]Department of Gastroenterology, The First Affiliated Hospital of Nanchang University, Jiangxi Province, China. [5]Laboratory of Molecular Pharmacology, Department of Pharmacology, School of Pharmacy, Southwest Medical University, Luzhou, China. [6]Department of Medicine, Howard University, Washington, DC, USA. [7]Cancer Center, Howard University, Washington, DC, USA. [8]Howard University Hospital, Howard University, Washington, DC, USA. [9]Department of Internal Medicine, Meharry Medical College, Nashville, TN, USA. [10]Lee Kong Chian School of Medicine, Nanyang Technological University, Singapore, Singapore. [11]Department of Medicine and Therapeutics, The Chinese University of Hong Kong, Hong Kong Special Administrative Region, Hong Kong, China. [12]Department of Gastroenterology, Shenzhen Hospital, Southern Medical University, Shenzhen, Guangdong, China. [13]The Third School of Clinical Medicine, Southern Medical University, Shenzhen, Guangdong, China. [14]State Key Laboratory of Cellular Stress Biology and School of Life Sciences, Xiamen University, Xiamen, China. [15]Institute for Microbial Ecology, School of Medicine, Department of Gastroenterology, Zhongshan Hospital, Xiamen University, Xiamen, China. [16]Department of Microbiology, The Chinese University of Hong Kong, Hong Kong Special Administrative Region, Hong Kong, China. [17]These authors contributed equally: Judeng Zeng, Chuan Xie. ✉e-mail: linzhang@cuhk.edu.hk; mtvchan@cuhk.edu.hk; huwei1683013@163.com; wukakei@cuhk.edu.hk

