## [Peer Review File · Nature Communications]

LOX-1 acts as an N6-methyladenosine-regulated receptor for *Helicobacter pylori* by binding to the bacterial catalaseEditorial Note: This manuscript has been previously reviewed at another journal that is not operating a transparent peer review scheme. This document only contains reviewer comments and rebuttal letters for versions considered at *Nature Communications*.

Reviewer #1 (Remarks to the Author):

All of my previous concerns have been addressed.

Reviewer #2 (Remarks to the Author):

1. This manuscript shows that *H. pylori* catalase interacts with host Lox-1. The authors correctly state there is an increase in antimicrobial resistance (Abstract). The authors hence suggest that anti-adhesive therapy might be of help (final sentence of abstract). However, as clearly shown in Fig. 5i the role of catalase in binding of *H. pylori* to human gastric tissue is highly variable, and hence, often, remaining bacteria will start growing again, leading to therapy failure. Hence this reviewer thinks that antiadhesive therapy via blocking Lox-1-catalase interaction is not the answer. For failure rates of any antimicrobial therapy it is crucial to know if therapy was given "blind" Or, alternatively, that antimicrobial sensitivity was determined first. The opinion of this reviewer is, that in areas where resistance is high, or when blind therapy fails, *H. pylori* should be grown from a gastric biopsy, and based on these outcomes, the correct antimicrobial therapy should be given. This is a more logical path to go as compared to giving anti-adhesive therapy, of which I do not know one single successful case. For influenza a few anti-adhesive compounds have passed the FDA. These compounds are extremely well acting in vitro, but their clinical effects are very, very small-no eradication of infection takes place but a small shortening of disease. It would be better when the authors remove their statements on clinical use from the manuscript.

2. The authors do quite a lot of experiments with mice (who do not express Lewis b, the major human receptor.) In addition they use the human gastric cell line HFE 145. To appreciate if blocking adhesion/entry of *H. pylori* with Lox-1 blockers in humans it is important to know if the *H. pylori* strains used express a function BabA (the major adhesin), and to know if HFE145 expresses Lewis b. In other words, can blocking Lox-1-catalase overrule binding due to BabA-Lewis b interaction.

Point-by-point response

Reviewers' comments

Reviewer #1 (Remarks to the Author):

All of my previous concerns have been addressed.

Response: We sincerely appreciate the reviewer's comments and positive response.

Reviewer #2 (Remarks to the Author):

Minor concern (1): 1. This manuscript shows that *H. pylori* catalase interacts with host Lox-1. The authors correctly state there is an increase in antimicrobial resistance (Abstract). The authors hence suggest that anti-adhesive therapy might be of help (final sentence of abstract). However, as clearly shown in Fig. 5i the role of catalase in binding of *H. pylori* to human gastric tissue is highly variable, and hence, often, remaining bacteria will start growing again, leading to therapy failure. Hence this reviewer thinks that antiadhesive therapy via blocking Lox-1-catalase interaction is not the answer. For failure rates of any antimicrobial therapy it is crucial to know if therapy was given "blind" Or, alternatively, that antimicrobial sensitivity was determined first. The opinion of this reviewer is, that in areas where resistance is high, or when blind therapy fails, *H. pylori* should be grown from a gastric biopsy, and based on these outcomes, the correct antimicrobial therapy should be given. This is a more logical path to go as compared to giving anti-adhesive therapy, of which I do not know one single successful case. For influenza a few anti-adhesive compounds have passed the FDA. These compounds are extremely well acting in vitro, but their clinical effects are very, very small-no eradication of infection takes place but a small shortening of disease. It would be better when the authors remove their statements on clinical use from the manuscript.

Response: We have added the related discussion (**Page 14-15, line 384-403**) to make it clear that blocking the interaction between LOX-1 and catalase could only reduce *H. pylori* adhesion and colonization to some extent but not completely.

Minor concern (2): The authors do quite a lot of experiments with mice (who do not express Lewis b, the major human receptor.) In addition they use the human gastric cell line HFE 145. To appreciate if blocking adhesion/entry of *H. pylori* with Lox-1 blockers in humans it is important to know if the *H. pylori* strains used express a function BabA (the major adhesin), and to know if HFE145 expresses Lewis b. In other words, can blocking Lox-1-catalase overrule binding due to BabA-Lewis b interaction.

Response: We have added **Supplementary Figure 12** to clarify the BabA and Lewis^b expression status of the bacteria strains and human cell line we used, respectively.